# Almost Sure Convergence of Stochastic Gradient Methods under Gradient Domination

**Simon Weissmann**[*]                                                                  *simon.weissmann@uni-mannheim.de*
*Institute of Mathematics*
*University of Mannheim*
*68138 Mannheim, Germany*

**Sara Klein**[*]                                                                          *sara.klein@uni-mannheim.de*
*Institute of Mathematics*
*University of Mannheim*
*68138 Mannheim, Germany*

**Waïss Azizian**                                                              *waiss.azizian@univ-grenoble-alpes.fr*
*Univ. Grenoble Alpes, CNRS*
*Inria, Grenoble INP, LJK*
*38000 Grenoble, France*

**Leif Döring**                                                                          *leif.doering@uni-mannheim.de*
*Institute of Mathematics*
*University of Mannheim*
*68138 Mannheim, Germany*

**Reviewed on OpenReview:** *https://openreview.net/forum?id=OTwnNBxZFB*

## Abstract

Stochastic gradient methods are among the most important algorithms in training machine learning problems. While classical assumptions such as strong convexity allow a simple analysis they are rarely satisfied in applications. In recent years, global and local gradient domination properties have shown to be a more realistic replacement of strong convexity. They were proved to hold in diverse settings such as (simple) policy gradient methods in reinforcement learning and training of deep neural networks with analytic activation functions. We prove almost sure convergence rates $f(X_n) - f^* \in o\big(n^{-\frac{1}{4\beta-1}+\epsilon}\big)$ of the last iterate for stochastic gradient descent (with and without momentum) under global and local $\beta$-gradient domination assumptions. The almost sure rates get arbitrarily close to recent rates in expectation. Finally, we demonstrate how to apply our results to the training task in both supervised and reinforcement learning.

## 1 Introduction

First-order methods to minimize an objective function $f$ have played a central role in the success of machine learning. This is accompanied by a growing interest in convergence statements particularly for stochastic gradient methods in different settings. To ensure convergence to the global optimum some kind of convexity assumption on the objective function is required. Especially in machine learning problems the standard

---

[*] These authors contributed equally to this work.

(strong) convexity assumption is nearly never fulfilled. However, it is well known that achieving convergence towards global optima is still possible under a weaker assumption, namely under the gradient domination property, often referred to as Polyak-Łojasiewicz (PL)-inequality (Polyak, 1963). Also in reinforcement learning, multiple results have shown that the objective function for policy gradient methods, under specific parametrizations, fulfills a weak type of gradient domination and therefore provably achieves convergence towards the global optimum (Mei et al., 2020; 2021; Fatkhullin et al., 2023; Klein et al., 2024). Improving the understanding of rates and optimal step size choices for stochastic first order methods is of significant interest for the machine learning and reinforcement learning community. Many classical results identify convergence rates for the expected error $\mathbb{E}[f(X_n) - f^*]$. In the present article, we focus on almost sure convergence rates for the error $f(X_n) - f^*$ in stochastic gradient schemes under weak gradient domination. The contribution of this work is as follows:

(i) Under global gradient domination with parameter $\beta$ (on the entire function domain), we prove that the last iterate of stochastic gradient descent (SGD) and stochastic heavy ball (SHB) converge almost surely and in expectation towards the global optimum with rate arbitrarily close to $o(n^{-\frac{1}{4\beta-1}})$. The almost sure and expectation rates of convergence that we obtain depend on the gradient domination parameter $\beta$ and are the same for both algorithms and convergence types. For SGD this rate is arbitrarily close to the tight upper bound known in expectation (Fatkhullin et al., 2022), while the almost sure convergence rate is new for the (weak) gradient domination assumption (see Theorem 4.1 and discussion afterwards). To the best of our knowledge for SHB this is the first convergence result towards global optima under (weak) gradient domination, for both almost sure convergence and convergence in expectation (see Theorem 4.2).

(ii) We consider the case where the gradient domination property holds only locally, either around stationary points or around global minima. We provide the first local convergence rates under these settings: we prove that SGD remains within the good local region with high probability and, conditioned on this event, we obtain converges rates almost surely and in expectation (see Theorem 5.1 and Theorem 5.2).

(iii) Our local setting covers generic classes of functions. In particular, we demonstrate that it encompasses the training task of deep neural networks with analytic activation functions in supervised learning. Our result illustrates that the iterates of SGD are likely to become trapped in areas of local minima when the step size is small. We verify under mild conditions, that SGD converges to local minima with given convergence speed (see Corollary 6.1).

(iv) Finally, we derive novel convergence results for policy gradient methods in reinforcement learning. We show that local gradient domination holds around the global optimum for the softmax parametrization resulting in the first local convergence rate for stochastic policy gradient with arbitrary batch-size (see Corollary 7.2).

We summarize the contributions of this paper in Table 1. These findings are also illustrated in a numerical toy experiment in Appendix B, where we have implemented SGD and SHB for monomials with increasing degree.

## 1.1 Literature Review and Classification of our Contribution

The roots of stochastic gradient methods trace back to Robbins and Monro (1951). Since then, various variants of SGD have been established as fundamental algorithms for optimizing complex models in the realm of machine learning. We refer to Bottou et al. (2018) and Garrigos and Gower (2024) for a detailed overview.

We start the review with the literature deriving convergence rates in expectation for SGD. Under the assumptions of smoothness and (strong) convexity Polyak (1987); Moulines and Bach (2011); Nguyen et al. (2018); Wang et al. (2021); Liu et al. (2023) studied convergence rates towards global optima. Moreover, many articles additionally analyze the non-convex case and prove convergence rates for the gradient norm towards zero (Ghadimi and Lan, 2013; Li et al., 2023; Liu et al., 2023; Nguyen et al., 2023).

Table 1: Summary of known and new results. Table presents convergence rates for tuned step size ($\epsilon > 0$ arbitrarily small). Dom.: gradient domination holds locally or globally; local*: additional assumption on $\gamma_1$ required and results holds only locally. a.s.: almost surely; $\mathbb{E}$: in expectation. Ref.: for some cited results minor adjustments are necessary.

| $\beta$ | Step size | Rate | Dom. | Algo. | Conv. | Ref. |
|---|---|---|---|---|---|---|
| $\frac{1}{2}$ | $\Theta\left(n^{-1+\epsilon}\right)$ | $o\left(n^{-1+\epsilon}\right)$ | global | SGD | a.s. | Theorem 4.1 (i); Liu and Yuan (2022, Thm. 1) |
| | | | | | $\mathbb{E}$ | Theorem 4.1 (ii); Khaled and Richtárik (2023, Thm. 3) |
| | | | | SHB | a.s. | Theorem 4.2 (i); Liu and Yuan (2022, Thm. 2) |
| | | | | | $\mathbb{E}$ | Theorem 4.2 (ii); Liang et al. (2023, Thm. 4.3) |
| | | | local* | SGD | a.s. | Theorem 5.1 (ii); Theorem 5.2 (ii) |
| | | | | | $\mathbb{E}$ | Theorem 5.1 (iii); Theorem 5.2 (iii); Mertikopoulos et al. (2020, Thm. 4) |
| $(\frac{1}{2}, 1]$ | $\Theta\left(n^{-\frac{2\beta}{4\beta-1}}\right)$ | $o\left(n^{-\frac{1}{4\beta-1}+\epsilon}\right)$ | global | SGD | a.s. | Theorem 4.1 (i) |
| | | | | | $\mathbb{E}$ | Theorem 4.1 (ii); Fatkhullin et al. (2022, Cor. 1) |
| | | | | SHB | a.s. | Theorem 4.2 (i) |
| | | | | | $\mathbb{E}$ | Theorem 4.2 (ii) |
| | | | local* | SGD | a.s. | Theorem 5.1 (ii); Theorem 5.2 (ii) |
| | | | | | $\mathbb{E}$ | Theorem 5.1 (iii); Theorem 5.2 (iii) |

Notably, several other results regarding convergence of SGD towards global optima have been established under the gradient domination setting (Karimi et al., 2016). Bassily et al. (2018); Liu et al. (2022) demonstrate exponential convergence rates in expectation in the overparameterized setting under strong gradient domination. See also Madden et al. (2024), where high-probability convergence bounds are shown using the (strong) gradient domination property. Further high-probability bounds on the approximation error are provided in Scaman et al. (2022) under a generalized gradient domination property, the so-called Separable-Łojasiewicz assumption, fulfilled by smooth neural networks. In Lei et al. (2020) also strong gradient domination is assumed, where the smoothness assumption is weakened through $\alpha$-Hölder continuity, achieving a rate of $O(\frac{1}{n^\alpha})$ in expectation. The Equation (ABC) condition is introduced in Khaled and Richtárik (2023) and where $O(\frac{1}{n})$ convergence is shown under strong gradient domination. Furthermore, Fatkhullin et al. (2022) and Fontaine et al. (2021) consider generalizations of gradient domination that include our definition as a special case. They derive convergence rates in expectation which we encompass with our result and extend to almost sure convergence (see also the discussion behind Theorem 4.1). Finally, Masiha et al. (2024) established both upper and lower bounds for projected gradient descent under gradient domination.

All the results mentioned so far consider convergence in expectation or high-probability bounds, although originally, motivated by Robbins and Siegmund (1971), research commenced with the quest for almost sure convergence rates for gradient methods. In recent years, Sebbouh et al. (2021) and, building upon it, Liu and Yuan (2022) derive almost sure convergence rates towards global optima under convexity for SGD and SHB. In addition, Liu and Yuan (2022) study SHB and Nesterov acceleration under PL. Returning the attention back to SGD with respect to gradient domination also some almost sure convergence results have been

established. As an extension to the PL-type gradient domination, in Chouzenoux et al. (2023) the so-called KL property is assumed, which contains gradient domination as a special case. The authors demonstrate almost sure convergence to a critical point, though without a rate. To conclude, to the best of our knowledge the derived almost sure convergence rate under gradient domination in Theorem 4.1 is novel.

Next, we want to provide further insights to the literature regarding SHB. In the realm of momentum methods, Polyak's Heavy-Ball Method (HBM) (Polyak, 1964) and Nesterov's accelerated gradient method (Nesterov, 1983) stand out as a foundational contribution. The authors of Gadat et al. (2018) provide a detailed description of the stochastic formulation of HBM and establish almost sure convergence but without giving a rate. In Yang et al. (2016); Orvieto et al. (2020); Yan et al. (2018); Mai and Johansson (2020); Zhou et al. (2020) convergence rates in expectation are shown in (strongly) convex and non-convex settings, where the non-convex analysis covers convergence of the norm of the gradient. Convergence of momentum methods under the strong gradient domination property and linear convergence due to an overparametrized machine learning setting is shown in Gess and Kassing (2023). Liang et al. (2023) determine $O(\frac{1}{n})$ convergence rate for SHB under strong gradient domination. Our main result for SHB presented in Theorem 4.2 describes almost sure convergence and convergence in expectation under global gradient domination. Both result are quantified with a given rate of convergence.

In the following, we aim to differentiate our local convergence analysis from existing results in the literature. There has been a lot of effort to derive local convergence guarantees for stochastic first order methods. In Dereich and Kassing (2024) almost sure convergence of SGD to a stationary point under the local gradient domination (for $x^*$) is demonstrated, provided that the process $(X_n)$ remains local, albeit without a rate. A local analysis of SGD towards minima without any gradient domination assumption is presented in Fehrman et al. (2020). Instead, a rank assumption is imposed on the Hessian, and mini-batches, along with resampling, are leveraged to ensure convergence to the global optimum with high probability. The resulting rate does not converge to zero and requires an increasing batch size. Finally, under the global Lipschitz assumption on the objective, it can be verified that SGD almost surely converges to a stationary point as demonstrated by Mertikopoulos et al. (2020). In the same work the authors derive a local convergence analysis under local strong convexity. Our analysis in Section 5 builds upon Mertikopoulos et al. (2020) and generalizes their results to the local gradient domination property. In our analysis we distinguish the cases where the local gradient domination property holds in a neighbourhood of a local minimum or in the neighbourhood of the global optimum respectively. Finally, we would like to acknowledge that related results have been independently obtained in the recent preprint Qiu et al. (2024).

For the application in the training of DNNs, it is worth noting that local convergence of SGD has been analyzed under stronger variants of gradient domination by Wojtowytsch (2023); An and Lu (2024). Due to the stronger form of gradient domination, specific sub-classes of DNNs need to be considered to verify these assumptions whereas our result is only constrained to analytic activation functions. Under the machine learning noise conditions in Wojtowytsch (2023), convergence toward zero loss with high probability is shown, provided that the initial loss is sufficiently small. In contrast, An and Lu (2024) demonstrate convergence towards zero loss under initialization in a local (strong) Łojasiewicz region. Indeed, one can construct DNNs satisfying the latter condition (Chatterjee, 2022).

For the application in reinforcement learning, recent results showed that choosing the tabular softmax parametrization in policy gradient (PG) algorithms results in objective functions which fulfill a non-uniform gradient domination property (Mei et al., 2020; Yuan et al., 2022; Klein et al., 2024). While convergence of PG for exact gradients is well understood, convergence rates for stochastic PG are rare and mostly require very large batch sizes (Ding et al., 2022; Klein et al., 2024; Ding et al., 2023). It is noteworthy that a similar local analysis for stochastic policy gradient under entropy regularization is presented in Ding et al. (2023). Their local result is also based on Mertikopoulos et al. (2020), but requires an increasing batch size sequence to obtain $O(\frac{1}{n})$-convergence towards the regularized optimum with high probability. In contrast, we consider both the unregularized and entropy regularized setting and observe that one can also achieve convergence arbitrarily close to $o(\frac{1}{n})$ without the need for an increasing batch size. Moreover, the local convergence occurs almost surely on an event with high probability.

## 2    Mathematical Background - Optimization under Gradient Domination

We consider the problem of solving the minimization problem of the form

$$\min_{x \in \mathbb{R}^d} f(x)\,, \tag{1}$$

where $f : \mathbb{R}^d \to \mathbb{R}$ denotes the objective function of interest. Throughout this paper we assume that the objective function is bounded from below by $f^* = \inf_{x \in \mathbb{R}^d} f(x) > -\infty$ and satisfies the classical L-smoothness assumption (either locally or globally):

**Assumption 2.1.** The objective function $f : \mathbb{R}^d \to \mathbb{R}$ is differentiable and the gradient $\nabla f$ is

(i)  globally $L$-Lipschitz continuous, i.e. there exists $L > 0$ such that $\|\nabla f(x) - \nabla f(y)\| \leq L\|x - y\|$ for all $x, y \in \mathbb{R}^d$.

(ii)  locally $L$-Lipschitz continuous, i.e. for all $R > 0$ there exists $L(R) > 0$ such that $\|\nabla f(x) - \nabla f(y)\| \leq L(R)\|x - y\|$ for all $x, y \in \mathbb{R}^d$ with $|x|, |y| \leq R$.

Using (global) $L$-smoothness, the descent lemma provides the inequality

$$f(y) \leq f(x) + \langle \nabla f(x), y - x \rangle + \frac{L}{2}\|y - x\|^2 \tag{2}$$

which is a fundamental instrument to analyze first order optimization methods. As a motivation recall the iterative update generated by gradient descent with constant step size $\gamma \leq \frac{1}{L}$, i.e.

$$x_{n+1} = x_n - \gamma \nabla f(x_n), \quad x_1 \in \mathbb{R}^d\,.$$

Applying Equation (2) and the iteration scheme yields the iterative descent property

$$[f(x_{n+1}) - f^*] \leq [f(x_n) - f^*] - \frac{\gamma}{2}\|\nabla f(x_n)\|^2\,.$$

Under further strong convexity assumption it is classical to show that the gradient descent algorithm converges to a global minimum at a linear rate. In order to derive a convergence rate without assuming convexity of $f$ one can use dominating relations of the gradient $\nabla f(x)$ with respect to the optimality gap $f(x) - f^*$. In particular, as demonstrated in Karimi et al. (2016) it is nowadays well-known that gradient descent converges linearly under the PL-condition which assumes that there exists $c > 0$ such that for all $x \in \mathbb{R}^d$ there holds

$$\|\nabla f(x)\| \geq c(f(x) - f^*)^\beta \tag{3}$$

with exponent $\beta = 1/2$ (Polyak, 1963). It is worthwhile to emphasize that Equation (3) is weaker than strong convexity, the classical textbook assumption that fails for many applications. Under the PL-condition the iterative descent property can be written as a recursion:

$$[f(x_{n+1}) - f^*] \leq \left(1 - \frac{\gamma c}{2}\right)[f(x_n) - f^*]\,.$$

In fact, there are many works analyzing (stochastic) first order methods under the weaker Łojasiewicz condition formulated in (local) areas around stationary points $x^*$ and exponents $\beta \in [1/2, 1]$ Lee et al. (2016); Fatkhullin et al. (2022); Scaman et al. (2022); Wilson et al. (2019). For general $\beta \in [1/2, 1]$ the recursive descent property reads as

$$[f(x_{n+1}) - f^*] \leq [f(x_n) - f^*] - \frac{\gamma c}{2}[f(x_n) - f^*]^{2\beta}$$

leading to sub-linear convergence for $\beta > 1/2$.

For the purpose of our analysis of stochastic gradient methods, we collect the following types of global and local gradient domination properties.

**Definition 2.2.** Let $f : \mathbb{R}^d \to \mathbb{R}$ be continuously differentiable with $f^* = \inf_{x \in \mathbb{R}^d} f(x) > -\infty$.

1. We say that $f$ satisfies the global gradient domination property with parameter $\beta \in [\frac{1}{2}, 1]$ if there exists $c > 0$ such that for all $x \in \mathbb{R}^d$ it holds true that

$$\|\nabla f(x)\| \geq c(f(x) - f^*)^\beta .$$

2. Let $x^* \in \mathbb{R}^d$ be a stationary point, i.e. $\nabla f(x^*) = 0$. We say that $f$ satisfies a local gradient domination property in $x^*$ with parameter $\beta_{x^*} \in [\frac{1}{2}, 1]$ if there exist a radius $r_{x^*} > 0$ and a constant $c_{x^*} > 0$ such that

$$\|\nabla f(x)\| \geq c_{x^*} |f(x) - f(x^*)|^{\beta_{x^*}} \tag{4}$$

for all $x \in \mathcal{B}_{r_{x^*}}(x^*) = \{y \in \mathbb{R}^d : \|x^* - y\| \leq r_{x^*}\}$. We say that $f$ satisfies a local gradient domination property in $f^*$ with parameter $\beta \in [\frac{1}{2}, 1]$ if there exist a radius $r > 0$ and a constant $c > 0$ such that

$$\|\nabla f(x)\| \geq c(f(x) - f^*)^\beta$$

for all $x \in \mathcal{B}_r^* = \{y \in \mathbb{R}^d : f(y) - f^* \leq r\}$.

*Remark* 2.3. If $\beta = \frac{1}{2}$, we will call the gradient domination *strong* since it is implied by strong convexity. In contrast, we call the gradient domination *weak* for $\beta \in (\frac{1}{2}, 1]$. Moreover, note that for the local gradient domination property in $x^*$ the parameters $r$ and $c$ may depend on $x^*$. Furthermore, we emphasize that for the definition of the local gradient domination in $f^*$ we do not require the existence of $x^* \in \arg\min_{x \in \mathbb{R}^d} f(x)$.

The PL-condition mentioned above is a special case of the general global gradient domination property for $\beta = \frac{1}{2}$. In Lojasiewicz (1965) it has been demonstrated that all analytic functions satisfy the local gradient domination property, emphasizing the particular significance of the local case. Further, it has been proved that all overparametrized neural networks fulfill the local gradient domination property (Liu et al., 2022). See also Madden et al. (2024); Dereich and Kassing (2024); Frei and Gu (2021) and references therein for the application of (strong) gradient domination to (deep) neural networks. In Fatkhullin et al. (2022); Attouch et al. (2010); Bolte et al. (2014); Zhou et al. (2018) examples of functions are discussed that fulfill the (weak) gradient domination property. For instance, one-dimensional monomials $f(x) = |x|^p$, $p \geq 2$, satisfy the weak global gradient domination property with $\beta = \frac{p-1}{p}$. We refer to Fatkhullin et al. (2022, Appendix A) for a longer list of globally gradient dominated functions including convex and non-convex functions. Notably, in reinforcement learning it is known that the tabular softmax parametrization leads to a parametrized value function that satisfies the so-called "non-uniform" PL-inequality (Mei et al., 2020; 2021; Klein et al., 2024). In Section 7 we will show how this non-uniform gradient domination implies local gradient domination in $f^*$. This renders our local analysis of stochastic gradient methods specifically applicable in RL. As mentioned earlier, since every analytic function already satisfies local gradient domination, we expect that the local analysis can encompass further parametrizations, such as neural networks.

## 2.1 Assumptions on the Stochastic First Order Oracle

Let $(\Omega, \mathcal{F}, \mathbb{P})$ be an underlying probability space. In general, we assume that we can access the exact gradient $\nabla f(x)$ through a stochastic first order oracle $V : \mathbb{R}^d \times M \to \mathbb{R}^d$ defined by

$$V(x, m) = \nabla f(x) + Z(x, m), \quad x \in \mathbb{R}^d, \ m \in M, \tag{5}$$

where $(M, \mathcal{M})$ is a measurable space, $Z : \mathbb{R}^d \times M \to \mathbb{R}^d$ is a state dependent $\mathcal{B}(\mathbb{R}^d) \otimes \mathcal{M}/\mathcal{B}(\mathbb{R}^d)$-measurable mapping describing the error to the exact gradient $\nabla f$. The stochastic gradient evaluation is then modelled through $V(x, \zeta)$, where the random variable $\zeta : \Omega \to M$ is independent of the state $x \in \mathbb{R}^d$. We make the following unbiasedness and second moment assumption:

**Assumption 2.4.** We assume that for each $x \in \mathbb{R}^d$ it holds that

$$\mathbb{E}[Z(x, \zeta)] := \int_\Omega Z(x, \zeta(\omega)) \mathrm{d}\mathbb{P}(\omega) = 0$$

and there exist non-negative constants $A, B$ and $C$ such that for all $x \in \mathbb{R}^d$ it holds that

$$\mathbb{E}[\|V(x, \zeta)\|^2] \leq A(f(x) - f^*) + B\|\nabla f(x)\|^2 + C . \tag{ABC}$$

It is worth noting that the Equation (ABC) assumption is a generalization of the bounded variance assumption that appears for $A = B = 0$. It was introduced by Khaled and Richtárik (2023) as expected smoothness condition and shown to be the weakest assumption among many others.

## 2.2 Stochastic Gradient Methods

The following two classical optimization algorithms will be analyzed in this article. Both algorithms are described as discrete time stochastic process $(X_n)_{n \in \mathbb{N}}$ driven by noisy gradient evaluations in Equation (5). In each iteration, we assume that the stochastic first order oracle is accessed through the evaluation of $\zeta_{n+1}$ which is a copy of $\zeta$ independent from the current state $X_n$.

The stochastic gradient descent (SGD) scheme is given by the stochastic update

$$X_{n+1} = X_n - \gamma_n V(X_n, \zeta_{n+1}),$$

where $X_1$ is a $\mathbb{R}^d$-valued random vector which denotes the initial state. To keep the notation simple, we will introduce $V_{n+1}(X_n) := V(X_n, \zeta_{n+1})$ suppressing the explicit noise representation through $(\zeta_n)_{n \in \mathbb{N}}$ in the following. The iterative update formula then reads as

$$X_{n+1} = X_n - \gamma_n V_{n+1}(X_n). \tag{SGD}$$

The iterative scheme of stochastic heavy ball (SHB) is defined by

$$X_{n+1} = X_n - \gamma_n V_{n+1}(X_n) + \nu(X_n - X_{n-1}), \tag{SHB}$$

with initial $\mathbb{R}^d$-valued random vector $X_1$. The additional summand is called the momentum term with momentum parameter $\nu \in [0, 1)$. In both cases, $(\gamma_n)_{n \in \mathbb{N}}$ denotes a sequence of positive step sizes and we denote by $(\mathcal{F}_n)_{n \in \mathbb{N}}$ the natural filtration induced by the process $(X_n)_{n \in \mathbb{N}}$. Note that we adopt the convention where the set of natural numbers $\mathbb{N}$ refers to the positive integers excluding zero. When necessary, we explicitly define $\mathbb{N}_0 := \mathbb{N} \cup \{0\}$.

*Example* 2.5 (Expected risk minimization). In order to give more insights into the considered setting of our stochastic first order oracle we formulate a specific one based on expected risk minimization. In expected risk minimization we are interested in minimizing an objective function of the form

$$f(x) = \mathbb{E}[F(x, \zeta)] = \int_\Omega F(x, \zeta(\omega)) \, d\mathbb{P}(\omega)$$

where $F : \mathbb{R}^d \times M \to \mathbb{R}$ is $\mathcal{B}(\mathbb{R}^d) \otimes \mathcal{M}/\mathcal{B}(\mathbb{R})$-measurable. In our notation the stochastic first order oracle then takes the form

$$V(x, \zeta) = \nabla f(x) + (\nabla_x F(x, \zeta) - \nabla f(x)) = \nabla_x F(x, \zeta)$$

and the iterative update of SGD reads as

$$X_{n+1} = X_n - \gamma_n \nabla_x F(X_n, \zeta_{n+1})$$

with a sequence of independent and identically distributed $(\zeta_n)$. The iterative scheme of SHB can be written in similar way. Note that this scenario also includes empirical risk minimization where the objective function takes a finite sum form, with $\zeta \sim \mathcal{U}(\{1, \ldots, N\})$,

$$f(x) = \frac{1}{N} \sum_{i=1}^N F(x, i) = \mathbb{E}[F(x, \zeta)].$$

Exemplifying the SGD method, we now illustrate the typical steps of the convergence analysis for first-order optimization methods. First, the smoothness of the function $f$ is exploited by applying the descent inequality Equation (2) to the iteration scheme and then applying conditional expectations,

$$\mathbb{E}[f(X_{n+1}) \mid \mathcal{F}_n] \le f(X_n) - \gamma_n \|\nabla f(X_n)\|^2 + \frac{L\gamma_n^2}{2} \mathbb{E}[\|V_{n+1}(X_n)\|^2 \mid \mathcal{F}_n].$$

Next, $f^*$ is subtracted on both sides and the variance term of the stochastic gradient is controlled through the Equation (ABC) condition:

$$\mathbb{E}[f(X_{n+1}) - f^* \mid \mathcal{F}_n] \leq \left(1 + \frac{LA\gamma_n^2}{2}\right)(f(X_n) - f^*) - \left(\gamma_n - \frac{BL\gamma_n^2}{2}\right)\|\nabla f(X_n)\|^2 + \frac{LC\gamma_n^2}{2}. \tag{6}$$

Without further assumptions this inequality can now be used to show that the gradient $\nabla f(X_n)$ converges almost surely to zero. In order to obtain convergence towards a global optimum additional assumptions are needed. For instance, it is sufficient to incorporate the global gradient domination property defined in Definition 2.2 which yields an iterative inequality of the form

$$\mathbb{E}[f(X_{n+1}) - f^* \mid \mathcal{F}_n] \leq \left(1 + \frac{LA\gamma_n^2}{2}\right)(f(X_n) - f^*) - \left(\gamma_n - \frac{BL\gamma_n^2}{2}\right)c^2(f(X_n) - f^*)^{2\beta} + \frac{LC\gamma_n^2}{2}. \tag{7}$$

Typically, the expectation is taken on both sides of the inequality to derive a convergence rate in expectation by working with recursive inequalities. In this article we push the argument further. We combine smoothness and gradient domination with a variant of the Robbins-Siegmund Theorem (see Lemma A.2) to derive almost sure convergence rates.

## 3 Preliminary Discussion on Super-Martingale Convergence Rates

In the previous section, we have sketched how to combine the global gradient domination property with smoothness to derive a recursive inequality of the form

$$\mathbb{E}[Y_{n+1} \mid \mathcal{F}_n] \leq (1 + c_1\gamma_n)Y_n - c_2\gamma_n Y_n^{2\beta} + c_3\gamma_n^2,$$

where $Y_n := f(X_n) - f^*$. For analysing these inequalities, we must deal separately with the strong gradient domination case ($\beta = \frac{1}{2}$) and the weak gradient domination case ($\beta > \frac{1}{2}$) to avoid divisions by zero. For the former case the recursive inequality simplifies, whereas a more complex analysis is required for the latter. To establish almost sure convergence rates we employ convergence lemmas for super-martingales based on the Robbins-Sigmund Theorem. This methodology has been introduced in Sebbouh et al. (2021) and further utilized in Liu and Yuan (2022) to analyze SGD and SHB under (strong) convexity. It is noteworthy to mention that their analysis is under the similar noise assumption formulated in Assumption 2.4.

In the following, we illustrate how to extend the arguments to convergence under the global gradient domination property. Here is our super-martingale result that also encompasses Liu and Yuan (2022, Lemma 1) when $\beta = \frac{1}{2}$ for completeness:

**Lemma 3.1.** *Let $(Y_n)_{n\in\mathbb{N}}$ be a sequence of non-negative random variables on an underlying probability space $(\Omega, \mathcal{F}, \mathbb{P})$ with natural filtration $(\mathcal{F}_n)_{n\in\mathbb{N}}$ and suppose there exists $\beta \in [\frac{1}{2}, 1]$, $c_1, c_3 \geq 0$ and $c_2 > 0$ such that*

$$\mathbb{E}[Y_{n+1} \mid \mathcal{F}_n] \leq (1 + c_1\gamma_n^2)Y_n - c_2\gamma_n Y_n^{2\beta} + c_3\gamma_n^2,$$

*for all $n \geq 1$, where $\gamma_n = \Theta(\frac{1}{n^\theta})$ for some fixed $\theta \in \left(\frac{1}{2}, 1\right)$. Then, for any*

$$\eta \in \begin{cases} \left(\max\{2 - 2\theta, \frac{\theta + 2\beta - 2}{2\beta - 1}\}, 1\right) & : \beta \in (\frac{1}{2}, 1] \\ (2 - 2\theta, 1) & : \beta = \frac{1}{2} \end{cases},$$

*$(Y_n)_{n\in\mathbb{N}}$ vanishes almost surely with $Y_n \in o\left(\frac{1}{n^{1-\eta}}\right)$.*

The proof of this lemma is based on the well-known Robbins-Siegmund theorem (Robbins and Siegmund, 1971, Theorem 1) and is provided in full detail in Appendix C.

## 4 Convergence for Global Gradient Domination Property

In this part of the paper, we provide our global convergence result in a non-convex and globally smooth setting. Combining the recursive inequality Equation (7) with the super-martingale convergence result from Lemma 3.1 leads to the following theorem.

**Theorem 4.1.** *Suppose Assumption 2.1 (i) and Assumption 2.4 are fulfilled and let $f$ satisfy the global gradient domination property from Definition 2.2 with $\beta \in [\frac{1}{2}, 1]$. Denote by $(X_n)_{n\in\mathbb{N}}$ the sequence generated by Equation* (SGD) *using a step size $\gamma_n = \Theta(\frac{1}{n^\theta})$ with $\theta \in (\frac{1}{2}, 1)$. For any*

$$\eta \in \begin{cases} \left(\max\{2 - 2\theta, \frac{\theta+2\beta-2}{2\beta-1}\}, 1\right) & : \beta \in (\frac{1}{2}, 1] \\ (2 - 2\theta, 1) & : \beta = \frac{1}{2} \end{cases}$$

*it holds that*

$$(i) \quad f(X_n) - f^* \in o\left(\frac{1}{n^{1-\eta}}\right), \quad a.s., \quad and \quad (ii) \quad \mathbb{E}[f(X_n) - f^*] \in o\left(\frac{1}{n^{1-\eta}}\right).$$

*Proof.* Recall, in Section 2 we derived Equation (7),

$$\mathbb{E}[f(X_{n+1}) - f^* \mid \mathcal{F}_n]$$
$$\leq \left(1 + \frac{LA\gamma_n^2}{2}\right)(f(X_n) - f^*) - \left(\gamma_n - \frac{BL\gamma_n^2}{2}\right)c^2(f(x) - f^*)^{2\beta} + \frac{LC\gamma_n^2}{2},$$

which will be the basis of the proof of Theorem 4.1.

We treat again both cases for $\beta = \frac{1}{2}$ and $\beta \in (\frac{1}{2}, 1]$ separately:

$\underline{\beta = \frac{1}{2}}$: In this case, Equation (7) results in the super-martingale inequality

$$\mathbb{E}[Y_{n+1} \mid \mathcal{F}_n] \leq \left(1 + \frac{LA\gamma_n^2}{2} - \gamma_n c^2 + \frac{BLc^2\gamma_n^2}{2}\right)Y_n + \frac{LC\gamma_n^2}{2},$$

with $Y_n = f(X_n) - f^*$. By the choice of $\gamma_n$ there exists $N > 0$ and a constant $\tilde{c} > 0$ such that $\gamma_n c^2 - \frac{LA\gamma_n^2}{2} - \frac{BLc^2\gamma_n^2}{2} \geq \tilde{c}\gamma_n$ for all $n \geq N$. Thus,

$$\mathbb{E}[Y_{n+1} \mid \mathcal{F}_n] \leq \left(1 - \tilde{c}\gamma_n\right)Y_n + \frac{LC\gamma_n^2}{2},$$

for all $n \geq N$. Then, claim (i) follows by applying Lemma 3.1 with $c_1 = 0, c_2 = \tilde{c}, c_3 = \frac{LC}{2}$ and $\beta = \frac{1}{2}$. To prove claim (ii) we multiply $(n+1)^{1-\eta}$ on both sides and take the expectation. It follows that

$$\mathbb{E}[(n+1)^{(1-\eta)}Y_{n+1}] \leq (1 - \tilde{c}\gamma_n)(n+1)^{1-\eta}\mathbb{E}[Y_n] + \frac{LC}{2}(n+1)^{1-\eta}\gamma_n^2$$

$$\leq (1 - \tilde{c}\gamma_n)(n^{1-\eta} + (1-\eta)n^{-\eta})\mathbb{E}[Y_n] + \frac{LC}{2}(n+1)^{1-\eta}\gamma_n^2$$

$$= \left(1 - \tilde{c}\gamma_n + \frac{1-\eta}{n} - \frac{\tilde{c}(1-\eta)\gamma_n}{n}\right)n^{1-\eta}\mathbb{E}[Y_n] + \frac{LC}{2}(n+1)^{1-\eta}\gamma_n^2.$$

As $\theta_n \in \Theta(\frac{1}{n^\theta})$ we obtain that $\tilde{c}\gamma_n$ is the dominating term. Hence, there exists a constant $\tilde{c}_1 > 0$ and $\tilde{N} > N$ such that $\tilde{c}\gamma_n - \frac{1-\eta}{n} + \frac{\tilde{c}(1-\eta)\gamma_n}{n} \geq \tilde{c}_1\gamma_n$ for all $n \geq \tilde{N}$. Thus, for all $n \geq \tilde{N}$

$$\mathbb{E}[(n+1)^{(1-\eta)}Y_{n+1}] \leq (1 - \tilde{c}_1\gamma_n)n^{1-\eta}\mathbb{E}[Y_n] + \frac{LC}{2}(n+1)^{1-\eta}\gamma_n^2.$$

We apply Lemma A.3 with $w_n = n^{1-\eta}\mathbb{E}[Y_n]$, $a_n = \tilde{c}_1\gamma_n$ and $b_n = (n+1)^{1-\eta}\gamma_n^2$ and obtain that $n^{1-\eta}\mathbb{E}[Y_n] \to 0$ for $n \to \infty$ which yields claim (ii). Note that $\sum_n b_n < \infty$ as $1 - \eta < 2\theta - 1$ for $\eta \in (2 - 2\theta, 1)$.

$\underline{\beta \in (\frac{1}{2}, 1]}$: In this case, Equation (7) results in the super-martingale inequality

$$\mathbb{E}[Y_{n+1} \mid \mathcal{F}_n] \leq \left(1 + \frac{LA\gamma_n^2}{2}\right)Y_n - \left(\gamma_n - \frac{BL\gamma_n^2}{2}\right)c^2 Y_n^{2\beta} + \frac{LC\gamma_n^2}{2},$$

with $Y_n = f(X_n) - f^*$. By the choice of $\gamma_n$ there exists $c_2 > 0$ and $N_1 > 0$ such that $c^2\gamma_n - \frac{BLc^2\gamma_n^2}{2} \geq c_2\gamma_n$ for all $n \geq N_1$,

$$\mathbb{E}[Y_{n+1} \mid \mathcal{F}_n] \leq \left(1 + \frac{LA\gamma_n^2}{2}\right)Y_n - c_2\gamma_n Y_n^{2\beta} + \frac{LC\gamma_n^2}{2}.$$

We deduce claim (i) from Lemma 3.1 with $c_1 = \frac{LA}{2}, c_2 = c_2, c_3 = \frac{LC}{2}$ and $\beta \in (\frac{1}{2}, 1]$.

For claim $(ii)$ we firstly proceed as in the proof of Lemma 3.1. Therefore, one can choose the auxiliary parameter $1 < q \leq \frac{1}{\theta}$ and find constants $c_4, c_3, \tilde{N}_1 > 0$ such that for all $n \geq \tilde{N}_1$ by EquationEquation (22) we have

$$\mathbb{E}[(n+1)^{1-\eta}Y_{n+1} \mid \mathcal{F}_n] \leq n^{1-\eta}Y_n - c_4\frac{1}{n^{q\theta}}n^{1-\eta}Y_n + c_3(n+1)^{1-\eta}(\gamma_n^{\frac{2\beta q-1}{2\beta-1}} + \gamma_n^2).$$

Next, we take the expectation to obtain

$$\mathbb{E}[(n+1)^{1-\eta}Y_{n+1}] \leq (1 - c_4\frac{1}{n^{q\theta}})\mathbb{E}[n^{1-\eta}Y_n] + c_3(n+1)^{1-\eta}(\gamma_n^{\frac{2\beta q-1}{2\beta-1}} + \gamma_n^2)$$

for all $n \geq \tilde{N}_1$, implying that $w_n = \mathbb{E}[n^{1-\eta}Y_n] \to 0$ as $n \to \infty$ by Lemma A.3. Note that we have chosen $\theta, \eta$ and $q$ as in Lemma 3.1, such that $\sum_n \frac{1}{n^{q\theta}} = \infty$, $\sum_n(n+1)^{1-\eta}\gamma_n^{\frac{2\beta q-1}{2\beta-1}} < \infty$, and $\sum_n(n+1)^{1-\eta}\gamma_n^2 < \infty$ (see Equation (23), Equation (24) and Equation (25)). Therefore, the assumptions of Lemma A.3 are met. $\square$

To the best of our knowledge our theorem presents the first convergence rate for SGD under weak gradient domination with respect to almost sure convergence.

It is natural to ask which $\theta$ leads to the best convergence rate. Optimizing for $\eta$ yields an optimal choice $\theta = \frac{2\beta}{4\beta-1}$ to achieve the best possible rate of convergence. This specific choice yields a lower bound of the interval given by $2 - 2\theta = \frac{\theta+2\beta-2}{2\beta-1} = 1 - \frac{1}{4\beta-1}$ and therefore an almost sure convergence of the form $o(\frac{1}{n^p})$ where $p$ is arbitrarily close to $\frac{1}{4\beta-1}$ (see also Table 1). We emphasize that the rate we obtain is arbitrarily close to the one obtained in Fontaine et al. (2021); Fatkhullin et al. (2022) in expectation. According to Fatkhullin et al. (2022, Prop. 2) the rate is attained, if the recursive inequality is indeed an equality.

Roughly speaking, our result guarantees a faster convergence rate for "stronger" gradient domination properties (i.e. for smaller $\beta$). Indeed, as $2 - 2\theta > \frac{\theta+2\beta-2}{2\beta-1}$ for $\beta$ sufficiently close to $\frac{1}{2}$ our result is consistent to the one presented in Liu and Yuan (2022, Thm. 1) by replacing the $\mu$-strongly convex assumption with the strong gradient domination property with $\beta = \frac{1}{2}$.

Similar arguments can be used to derive almost sure convergence rates for SHB under global gradient domination:

**Theorem 4.2.** *Suppose Assumption 2.1 (i) and Assumption 2.4 are fulfilled and let $f$ satisfy the global gradient domination property from Definition 2.2 with $\beta \in [\frac{1}{2}, 1]$. Denote by $(X_n)_{n\in\mathbb{N}}$ the sequence generated by Equation (SHB) using a step size $\gamma_n = \Theta(\frac{1}{n^\theta})$ for $\theta \in (\frac{1}{2}, 1)$. For any*

$$\eta \in \begin{cases} \left(\max\{2 - 2\theta, \frac{\theta+2\beta-2}{2\beta-1}\}, 1\right) & : \beta \in (\frac{1}{2}, 1] \\ (2 - 2\theta, 1) & : \beta = \frac{1}{2} \end{cases}$$

*it holds that*

$$(i) \quad f(X_n) - f^* \in o\left(\frac{1}{n^{1-\eta}}\right), \quad a.s., \quad and \quad (ii) \quad \mathbb{E}[f(X_n) - f^*] \in o\left(\frac{1}{n^{1-\eta}}\right).$$

For the proof of Theorem 4.2, recall the definition of the iteration scheme Equation (SHB). Using the following definitions

$$Z_n = X_n + \frac{\nu}{1-\nu}W_n, \quad W_n = X_n - X_{n-1}, \tag{8}$$

one can derive the iterative evolution

$$W_{n+1} = \nu W_n - \gamma_n V(X_n) \tag{9}$$

$$Z_{n+1} = Z_n - \frac{\gamma_n}{1-\nu} V(X_n). \tag{10}$$

The first equation follows directly from the definition of $W_n$. For the second equation we compute

$$
\begin{aligned}
Z_{n+1} = X_{n+1} + \frac{\nu}{1+\nu} W_{n+1} &= (1 + \frac{\nu}{1-\nu}) X_{n+1} - \frac{\nu}{1-\nu} X_n \\
&= \frac{1-\nu}{1-\nu} X_n - \frac{\gamma_n}{1-\nu} V(X_n) + \frac{\nu}{1-\nu} (X_n - X_{n-1}) \\
&= Z_n - \frac{\gamma_n}{1-\nu} V(X_n).
\end{aligned}
$$

We will utilize these auxiliary variables in the proof.

*Proof of Theorem 4.2.* The proof begins as in the proof of Theorem 2 in Liu and Yuan (2022). Using only $L$-smoothness and assumption Equation (ABC), they show that for any $c_3 \in (0, \frac{1}{1-\nu})$, $\lambda \in (\nu, 1)$ there exist constants $c_1, c_2, c_4 > 0$ such that choosing the step size $\gamma_n \sim \frac{1}{n^\theta}$, for some $\theta \in (\frac{1}{2}, 1)$ results in (Liu and Yuan, 2022, Equation (21))

$$
\begin{aligned}
&\mathbb{E}[f(Z_{n+1}) - f^* + \|W_{n+1}\|^2 \mid \mathcal{F}_n] \\
&\le (1 + c_1 \gamma_n^2)(f(Z_n) - f^*) + (\lambda + c_2 \gamma_n^2)\|W_n\|^2 - c_3 \gamma_n \|\nabla f(Z_n)\|^2 + c_4 \gamma_n^2
\end{aligned}
\tag{11}
$$

for all $n \ge N$ and some $N > 0$ sufficiently large. Next, we apply the global gradient domination property for any $\beta \in [\frac{1}{2}, 1]$ to derive

$$
\begin{aligned}
&\mathbb{E}[f(Z_{n+1}) - f^* + \|W_{n+1}\|^2 \mid \mathcal{F}_n] \\
&\le (1 + c_1 \gamma_n^2)(f(Z_n) - f^*) - cc_3 \gamma_n (f(Z_n) - f^*)^{2\beta} + (\lambda + c_2 \gamma_n^2)\|W_n\|^2 + c_4 \gamma_n^2.
\end{aligned}
\tag{12}
$$

For the remaining proof, we denote $Q_n := f(Z_n) - f^*$. Similar as before, we treat both cases for $\beta = \frac{1}{2}$ and $\beta \in (\frac{1}{2}, 1]$ separately:

$\beta = \frac{1}{2}$: Instead of $\mu$-strong convexity we use the gradient domination inequality $\|\nabla f(x)\|^2 \ge c(f^* - f(x))$, as the same inequality is implied by strong convexity using $c = \mu$. Then, Claim (i) follows using the same proof as Liu and Yuan (2022, Thm. 2b)). Note that the inequality

$$\frac{1}{2L} \|\nabla f(x)\|^2 \le f(x) - f^*, \tag{13}$$

used in the last step only requires the $L$-smoothness assumption (Nesterov, 2004, Sec. 1.2.3).

For Claim (ii) we consider Equation (12) which simplifies for $\beta = \frac{1}{2}$ to

$$\mathbb{E}[Q_{n+1} + \|W_{n+1}\|^2 \mid \mathcal{F}_n] \le (1 + c_1 \gamma_n^2 - cc_3 \gamma_n) Q_n + (\lambda + c_2 \gamma_n^2)\|W_n\|^2 + c_4 \gamma_n^2.$$

By the choice of $\gamma_n$ there exists $N > 0$ and $\tilde{c}_1, \tilde{c}_2 > 0$, such that $cc_3 \gamma_n - c_1 \gamma_n^2 \ge \tilde{c}_1 \gamma_n$ and $\lambda + c_2 \gamma_n^2 \le \tilde{c}_2 \gamma_n$ for all $n \ge N$. Hence, for $n \ge N$

$$
\begin{aligned}
\mathbb{E}[Q_{n+1} + \|W_{n+1}\|^2 \mid \mathcal{F}_n] &\le (1 - \tilde{c}_1 \gamma_n) Q_n + (1 - \tilde{c}_2 \gamma_n)\|W_n\|^2 + c_4 \gamma_n^2 \\
&\le (1 - \min\{\tilde{c}_1, \tilde{c}_2\}) \left(Q_n + \|W_n\|^2\right) + c_4 \gamma_n^2.
\end{aligned}
$$

Let $c_5 = \min\{\tilde{c}_1, \tilde{c}_2\}$, multiply by $(n+1)^{1-\eta}$ on both sides and use Equation (17) to obtain for $n \ge N$

$$
\begin{aligned}
&\mathbb{E}[(n+1)^{1-\eta} \left(Q_{n+1} + \|W_{n+1}\|^2\right) \mid \mathcal{F}_n] \\
&\le (n+1)^{1-\eta}(1 - c_5 \gamma_n) \left(Q_n + \|W_n\|^2\right) + c_4 \gamma_n^2 (n+1)^{1-\eta} \\
&\le (n^{1-\eta} + (1-\eta)n^{-\eta})(1 - c_5) \left(Q_n + \|W_n\|^2\right) + c_4 \gamma_n^2 (n+1)^{1-\eta} \\
&= \left(1 - c_5 \gamma_n + \frac{1-\eta}{n} - \frac{c_5(1-\eta)\gamma_n}{n}\right) n^{1-\eta} \left(Q_n + \|W_n\|^2\right) + c_4 \gamma_n^2 (n+1)^{1-\eta}.
\end{aligned}
$$

Taking expectation and using that there exists $\tilde{c}_5 > 0$ and $\tilde{N} > N$ such that $c_5\gamma_n - \frac{1-\eta}{n} + \frac{c_5(1-\eta)\gamma_n}{n} \geq \tilde{c}_5\gamma_n$, we have for all $n \geq \tilde{N}$

$$\mathbb{E}[(n+1)^{1-\eta}\left(Q_{n+1} + \|W_{n+1}\|^2\right)] \leq (1 - \tilde{c}_5\gamma_n)\,\mathbb{E}\left[n^{1-\eta}\left(Q_n + \|W_n\|^2\right)\right] + c_4\gamma_n^2(n+1)^{1-\eta}.$$

Note that $\sum_n \gamma_n^2(n+1)^{1-\eta} < \infty$ because $\eta \in (2 - 2\theta, 1)$ implies $1 - \eta < 2\theta - 1$. We can apply Lemma A.3 which yields that $\mathbb{E}\left[n^{1-\eta}\left(Q_n + \|W_n\|^2\right)\right] \to 0$. Hence, $\mathbb{E}[(Q_n + \|W_n\|^2)] \in o(\frac{1}{n^{1-\eta}})$.
To finish the proof, one can derive

$$f(X_n) - f^* \leq Q_n + \frac{1}{2}\|\nabla f(Z_n)\|^2 + \frac{\nu^2 + L\nu^2}{2(1-\nu)^2}\|W_n\|^2. \tag{14}$$

To derive this equation, recall that $Z_n - X_n = -\frac{\nu}{1-\nu}W_n$, such that by $L$-smoothness we obtain

$$f(X_n) \leq f(Z_n) + \langle \nabla f(Z_n), X_n - Z_n \rangle + \frac{L}{2}\|X_n - Z_n\|^2 = f(Z_n) - \frac{\nu}{1-\nu}\langle \nabla f(Z_n), W_n \rangle + \frac{L\nu^2}{2(1-\nu)^2}\|W_n\|^2.$$

Next, apply Cauchy-Schwarz and Young's inequality to obtain

$$f(X_n) - f^* \leq f(Z_n) - f^* + \frac{1}{2}\|\nabla f(Z_n)\|^2 + \frac{\nu^2}{2(1-\nu)^2}\|W_n\|^2 + \frac{L\nu^2}{2(1-\nu)^2}\|W_n\|^2.$$

Using inequality Equation (13), we get almost surely

$$f(X_n) - f^* \leq (1+L)Q_n + \frac{\nu^2 + L\nu^2}{2(1-\nu)^2}\|W_n\|^2 \leq (1+L)\max\left(1, \frac{\nu^2}{2(1-\nu)^2}\right)(Q_n + \|W_n\|^2). \tag{15}$$

implying that $\mathbb{E}[(f(X_n) - f^*)] \in o(\frac{1}{n^{1-\eta}})$ which proves Claim (ii).

$\underline{\beta \in (\frac{1}{2}, 1]}$: For Claim (i), note that in Equation (12) $\lambda < 1$, such that

$$\mathbb{E}[Q_{n+1} + \|W_{n+1}\|^2 \mid \mathcal{F}_n] \leq (1 + c_1\gamma_n^2)Q_n + (1 + c_2\gamma_n^2)\|W_n\|^2 + cc_3\gamma_n Q_n^{2\beta} + c_4\gamma_n^2$$
$$\leq (1 + \max\{c_1, c_2\}\gamma_n^2)(Q_n + \|W_n\|^2) + cc_3\gamma_n(Q_n + \|W_n\|^2)^{2\beta} + c_4\gamma_n^2.$$

By Lemma 3.1 we obtain that $Q_n + \|W_n\|^2 = f(Z_n) - f^* + \|W_n\|^2 \in o\left(\frac{1}{n^{1-\eta}}\right)$ for all $\eta \in \left(\max\{2 - 2\theta, \frac{\theta + 2\beta - 2}{2\beta - 1}\}, 1\right)$. We apply the inequality in Equation (15) to conclude that also $f(X_n) - f^* \in o\left(\frac{1}{n^{1-\eta}}\right)$ for all $\eta \in \left(\max\{2 - 2\theta, \frac{\theta + 2\beta - 2}{2\beta - 1}\}, 1\right)$. This proves Claim (i).

For Claim (ii), we again use the $q$-trick from Lemma 3.1 in Equation (12). For $1 < q < \frac{1}{\theta} < 2$ we have that

$$\mathbb{E}[Q_{n+1} + \|W_{n+1}\|^2 \mid \mathcal{F}_n] \leq (1 + c_1\gamma_n^2 - cc_3\gamma_n^q)Q_n + cc_3\gamma_n\left(\gamma_n^{q-1}Q_n - Q_n^{2\beta}\right) + (\lambda + c_2\gamma_n^2)\|W_n\|^2 + c_4\gamma_n^2.$$

Now with Equation (20) in Lemma 3.1 there exists $\tilde{c}_3 \geq 0$ such that

$$\mathbb{E}[Q_{n+1} + \|W_{n+1}\|^2 \mid \mathcal{F}_n] \leq (1 + c_1\gamma_n^2 - cc_3\gamma_n^q)Q_n + \tilde{c}_3\gamma_n^{\frac{2\beta q - 1}{2\beta - 1}} + (\lambda + c_2\gamma_n^2)\|W_n\|^2 + c_4\gamma_n^2.$$

By the choice of $\gamma_n$ there exists $\tilde{c}_1 > 0$ and $N > 0$ such that $c_1\gamma_n^2 - cc_3\gamma_n^q \geq \tilde{c}_1\gamma_n^q$ and $\lambda + c_2\gamma_n^2 \leq \tilde{c}_1\gamma_n^q$ for all $n \geq N$. Thus, for all $n \geq N$,

$$\mathbb{E}[Q_{n+1} + \|W_{n+1}\|^2 \mid \mathcal{F}_n] \leq (1 - \tilde{c}_1\gamma_n^q)(Q_n + \|W_n\|^2) + \max\{\tilde{c}_3, c_4\}\left(\gamma_n^{\frac{2\beta q - 1}{2\beta - 1}} + \gamma_n^2\right).$$

For $\max\{\tilde{c}_3, c_4\} =: \tilde{c}_2$, we multiply on both sides with $(n+1)^{1-\eta}$ and take the expectation to obtain for $n \geq N$

$$
\mathbb{E}[(n+1)^{1-\eta} \left(Q_{n+1} + \|W_{n+1}\|^2\right)]
$$

$$
\leq (n+1)^{1-\eta}(1 - \tilde{c}_1 \gamma_n^q)\mathbb{E}[(Q_n + \|W_n\|^2)] + \tilde{c}_2(n+1)^{1-\eta} \left(\gamma_n^{\frac{2\beta q - 1}{2\beta - 1}} + \gamma_n^2\right)
$$

$$
\leq (n^{1-\eta} + (1-\eta)n^{-\eta})(1 - \tilde{c}_1 \gamma_n^q)\mathbb{E}[(Q_n + \|W_n\|^2)] + \tilde{c}_2(n+1)^{1-\eta} \left(\gamma_n^{\frac{2\beta q - 1}{2\beta - 1}} + \gamma_n^2\right)
$$

$$
= \left(1 - \tilde{c}_1 \gamma_n^q + \frac{1-\eta}{n} - \frac{\tilde{c}_1(1-\eta)\gamma_n^q}{n}\right) \mathbb{E}[n^{1-\eta}(Q_n + \|W_n\|^2)]
$$

$$
+ \tilde{c}_2(n+1)^{1-\eta} \left(\gamma_n^{\frac{2\beta q - 1}{2\beta - 1}} + \gamma_n^2\right).
$$

Next, there exists $\tilde{N} > N$ and $\tilde{c}_5 > 0$ such that for all $n \geq \tilde{N}$

$$
\mathbb{E}[(n+1)^{1-\eta} \left(Q_{n+1} + \|W_{n+1}\|^2\right)] \leq (1 - \tilde{c}_5 \gamma_n^q) \mathbb{E}[n^{1-\eta}(Q_n + \|W_n\|^2)] + \tilde{c}_2(n+1)^{1-\eta} \left(\gamma_n^{\frac{2\beta q - 1}{2\beta - 1}} + \gamma_n^2\right).
$$

From the proof of Lemma 3.1, we choose the auxiliary parameter $q$ such that $\sum_n (n+1)^{1-\eta} \left(\gamma_n^{\frac{2\beta q - 1}{2\beta - 1}} + \gamma_n^2\right) < \infty$ (see Equation (24) and Equation (25)). By applying again Lemma A.3 we obtain $\mathbb{E}[n^{1-\eta}(Q_n + \|W_n\|^2)] \to 0$, i.e. $\mathbb{E}[Q_n + \|W_n\|^2] \in o(\frac{1}{n^{1-\eta}})$. Finally, Claim (ii) follows again by Equation (15). $\square$

To the best of our knowledge, our result gives the first convergence proof of SHB to global optima under weak gradient domination, with rates for almost sure convergence and convergence of expectations. The resulting convergence rate using the optimized step size are summarized in Table 1. In the strong gradient domination setting our rate in expectation gets arbitrarily close to the $O(\frac{1}{n})$ convergence obtained in Liang et al. (2023). It is noteworthy that the utilization of SHB in our analysis does not yield a superior convergence rate compared to SGD. This arises from the proof technique and aligns with the findings in Liu and Yuan (2022); Sebbouh et al. (2021) where the authors similarly achieve no acceleration. In general, for deterministic settings acceleration of gradient methods can achieve improvements of convergence rates (Wilson et al., 2019). Though in the special case of gradient domination with $\beta = \frac{1}{2}$, HB as well as Nesterov cannot accelerate in the deterministic setting as shown in Yue et al. (2023).

## 5 Convergence for Local Gradient Domination Property

In this section, we want to generalize the analysis in Mertikopoulos et al. (2020) under local strong convexity to the weaker local gradient domination property for different cases of $\beta$. We consider the two cases of local gradient domination separately. The contributions and differences of our results under less restricted assumptions are the following:
First, we show in both cases that SGD remains in the gradient dominated region with high probability by only assuming local gradient domination instead of local strong convexity. Especially in the case of a local minimum $x^*$ this is a challenging task, as we have to ensure that the SGD scheme $(X_n)_{n \in \mathbb{N}}$ remains close to $x^*$ without exploiting convexity. We can guarantee this whenever $x^*$ is in an isolated connected compact set of local minima $\mathcal{X}^*$. We prove convergence towards the level set of $\mathcal{X}^*$ and obtain Theorem 5.1. Second, additionally to convergence in expectation we prove almost sure convergence conditioned on the "good event". Third, due to the weaker gradient domination assumption and no convexity, one cannot expect the convergence of $X_n$ to (local or global) minimum $x^*$, instead we focus on convergence of $f(X_n)$ to $f(x^*)$. In Liu and Zhou (2023) they delve into the rationale behind considering this as a more robust metric.

The main result under local gradient domination in a local minimum $x^*$ is as follows:

**Theorem 5.1.** *Fix some tolerance level $\delta > 0$ and let $\mathcal{X}^* \subset \mathbb{R}^d$ be an isolated compact connected set of local minima with level $l = f(x^*)$ for all $x^* \in \mathcal{X}^*$. Suppose that $f$ satisfy the local gradient domination*

*property in each $x^* \in \mathcal{X}^*$, $f$ is locally $G$-Lipschitz continuous and satisfies Assumption 2.1 (ii). Moreover, suppose Assumption 2.4 hold true. Denote by $(X_n)_{n \in \mathbb{N}}$ the sequence generated by Equation* (SGD) *using a step size $\gamma_n = \Theta(\frac{1}{n^\theta})$ for $\theta \in (\frac{1}{2}, 1)$ and suppose that $\gamma_n \le \gamma_1$ for $\gamma_1$ sufficiently small enough such that $\sum_{n=1}^\infty \gamma_n^2 < \frac{\delta\epsilon}{2(G^2C^2+G^2+C)}$ for some $\epsilon > 0$ independent of $\delta$. Then, the following holds:*

(i) *There exist subsets $\mathcal{U}$ and $\mathcal{U}_1$ of $\mathbb{R}^d$ such that, if $X_1 \in \mathcal{U}_1$ the event $\Omega_{\mathcal{U}} = \{X_n \in \mathcal{U} \text{ for all } n = 1, 2, \dots\}$ has probability at least $1 - \delta$.*

*Moreover, there exists $\beta \in [\frac{1}{2}, 1]$ such that for any*

$$\eta \in \begin{cases} \left(\max\{2 - 2\theta, \frac{\theta + 2\beta - 2}{2\beta - 1}\}, 1\right) & : \beta \in (\frac{1}{2}, 1] \\ (2 - 2\theta, 1) & : \beta = \frac{1}{2} \end{cases}$$

*it holds that*

(ii) $|f(X_n) - l|\mathbf{1}_{\Omega_{\mathcal{U}}} \in o\left(\frac{1}{n^{1-\eta}}\right)$, *a.s.,* *and* (iii) $\mathbb{E}[|f(X_n) - l|\mathbf{1}_{\Omega_{\mathcal{U}}}] \in o\left(\frac{1}{n^{1-\eta}}\right)$.

We would like to emphasize that our result does not provide convergence rates in high probability. Instead, we restrict the probability space to "good" events (encoded in $\mathcal{U}$) that occur with high probability $1 - \delta$. Conditioned on these events, we establish convergence with a rate that is independent of $\delta$. The dependence on $\delta$ instead manifests in the step size $\gamma_n$, which ensure that the iterations $(X_n)_{n \in \mathbb{N}}$ remain within $\mathcal{U}$ with probability at least $\delta$. Notably, this is the only aspect of our convergence result that depends on $\delta$.

In the following, we only sketch the proof in the main part and provide the full proof in Appendix D.

*Sketch of proof.* The proof is split into several intermediate steps outlined as follows:

- First, we unify the gradient domination property around the set of local minima $\mathcal{X}^*$ and obtain a radius $\mathbf{r}$ such that the unified gradient domination property is fulfilled in all open balls with radius $\mathbf{r}$ around $x^* \in \mathcal{X}^*$ (Lemma D.1).

- Based on this we construct the two sets $\mathcal{U}, \mathcal{U}_1 \subseteq \mathbb{R}^d$ defined as neighborhoods of $\mathcal{X}^*$ constructed such that the gradient domination property holds within this region,

$$\mathcal{U}_1 = \{x \in \mathbb{R}^d : \inf_{x^* \in \mathcal{X}^*} \|x - x^*\| < \frac{\mathbf{r}}{2}, f(x) - l \le \frac{\epsilon}{2}\}$$
$$\mathcal{U} = \{x \in \mathbb{R}^d : \inf_{x^* \in \mathcal{X}^*} \|x - x^*\| < \frac{\mathbf{r}}{2}\}$$

  and the events

$$\Omega_n = \{X_k \in \mathcal{U} \text{ for all } k \le n\} \in \Omega$$

  such that $\Omega_{\mathcal{U}} = \bigcap_n \Omega_n$ occurs with high probability. This means, when starting in $\mathcal{U}_1$ the gradient trajectory does remain in $\mathcal{U}$ for all gradient steps with high probability.

- We present a row of Lemmata to show that $\mathbb{P}(\Omega_n) \ge 1 - \delta$ for all $n \in \mathbb{N}$, claim (i) of the Theorem. To show $\mathbb{P}(\Omega_n) \ge 1 - \delta$ we construct the sets $C_n$ and $E_n$ defined in Equation (29) and Equation (34) such that $E_n \cap C_n \subset \Omega_{n+1}$ (Lemma D.6) while Lemma D.5 is used to prove this claim. The sets $E_n$ are such that $f(X_n)$ remains close to $f^*$. We exploit the unified gradient domination property to construct the sets $E_n$ (Lemma D.4) and derive a recursive inequality in Lemma D.6 c) to prove that this event occurs with high probability (Lemma D.7). The sets $C_n$ are such that $X_{n+1}$ remains close to $X_n$ and we exploit the finite variance assumption to show that these events occur with high probability (Lemma D.8).

- Finally, claim (ii) and (iii) are shown directly in the proof of Theorem 5.1 at the end of Appendix D, where we employ the uniform gradient domination in $\Omega_{\mathcal{U}}$.

$\square$

The main result concerning local gradient domination in $f^*$ is presented below and does not necessitate the existence of a local minimum or any stationary point. It is worth noting that the definition of local gradient domination in $f^*$ guarantees the gradient domination property for any $x$ with $f(x)$ close to $f^*$. Consequently, this definition ensures that functions satisfying this property cannot possess local minima or saddle points within this region.

**Theorem 5.2.** *Fix some tolerance level $\delta > 0$. Suppose $f$ satisfies the local gradient domination property in $f^*$ from Definition 2.2 with $\beta \in [\frac{1}{2}, 1]$ and $\mathcal{B}_r^* \subseteq \mathbb{R}^d$. Moreover, suppose within $\mathcal{B}_r^*$ $f$ is $G$-Lipschitz continuous, Assumption 2.1 (i) and Assumption 2.4 hold true. Denote by $(X_n)_{n \in \mathbb{N}}$ the sequence generated by Equation* (SGD) *using a step size $\gamma_n = \Theta(\frac{1}{n^\theta})$ for $\theta \in (\frac{1}{2}, 1)$ and suppose that $\gamma_n \leq \gamma_1$ for $\gamma_1$ sufficiently small such that $\sum_{n=1}^{\infty} \gamma_n^2 < \frac{\delta \epsilon}{2(G^2 C^2 + G^2 + C)}$ for some $\epsilon > 0$ independent of $\delta$. Then, the following holds:*

*(i) There exist subsets $\mathcal{U}$ and $\mathcal{U}_1$ of $\mathbb{R}^d$ such that, if $X_1 \in \mathcal{U}_1$ the event $\Omega_{\mathcal{U}} = \{X_n \in \mathcal{U} \text{ for all } n = 1, 2, \dots\}$ has probability at least $1 - \delta$.*

*Moreover, for any*

$$\eta \in \begin{cases} \left(\max\{2 - 2\theta, \frac{\theta + 2\beta - 2}{2\beta - 1}\}, 1\right) & : \beta \in (\frac{1}{2}, 1] \\ (2 - 2\theta, 1) & : \beta = \frac{1}{2} \end{cases}$$

*it holds that*

$$(ii) \quad (f(X_n) - f^*)\mathbf{1}_{\Omega_{\mathcal{U}}} \in o\left(\frac{1}{n^{1-\eta}}\right), \quad a.s., \quad and \quad (iii) \quad \mathbb{E}[(f(X_n) - f^*)\mathbf{1}_{\Omega_{\mathcal{U}}}] \in o\left(\frac{1}{n^{1-\eta}}\right).$$

The outline of the proof is similar to the proof of Theorem 5.1. We provide the full proof and more details on the sets $\mathcal{U}$ and $\mathcal{U}_1$ as well as on $\beta$ in Appendix D.

# 6   Application in the training of neural networks

In supervised learning one aims to approximate an unknown model $\varphi : \mathbb{R}^{d_z} \to \mathbb{R}^{d_y}$ by a parametrized function $g_w : \mathbb{R}^{d_z} \to \mathbb{R}^{d_y}$ with parameter $w \in \mathbb{R}^{d_w}$. Given a family of training data $((Z^{(m)}, Y^{(m)}))_{m \in \mathbb{N}}$ generated as i.i.d. samples from an unknown distribution $\mu_{(Z,Y)}$ one usually chooses the parameter $w \in \mathbb{R}^{d_w}$ by solving

$$\min_{w \in \mathbb{R}^{d_w}} \mathbb{E}_{\mu_{(Z,Y)}}[\Phi(g_w(Z), Y)],$$

where $\Phi : \mathbb{R}^{d_y} \times \mathbb{R}^{d_y} \to \mathbb{R}_+$ is a user specific data discrepancy. One popular choice of parametrizations are DNNs. We define a neural network of depth $L \in \mathbb{N}$ by the recursion

$$z_0 := z, \quad z_\ell = \sigma^{\otimes d_\ell}(A_\ell z_{\ell-1} + b_\ell), \ \ell = 1, \dots, L-1, \quad g_w(z) := A_L z_{L-1} + b_L.$$

The weights $((A_\ell, b_\ell))_{\ell=1}^L$ of the DNN are collected in $w \in \mathcal{W} := \times_{\ell=1}^L (\mathbb{R}^{d_\ell \times d_{\ell-1}} \times \mathbb{R}^{d_\ell}) \simeq \mathbb{R}^{d_w}$, and $\sigma^{\otimes d} : \mathbb{R}^d \to \mathbb{R}^d$ describes the component-wise application of the activation function $\sigma : \mathbb{R} \to \mathbb{R}$.

Provided that $\sigma$ and $\Phi$ are analytic, and $(Z, Y)$ are compactly supported $\mathbb{R}^{d_z} \times \mathbb{R}^{d_y}$-valued random variables, then $f^{\mathrm{DNN}} : \mathbb{R}^{d_w} \to \mathbb{R}_+$ defined by $w \mapsto \mathbb{E}_{\mu_{(Z,Y)}}[\Phi(g_w(Z), Y)]$ is analytic (Dereich and Kassing, 2024, Thm. 5.2). By Lojasiewicz (1963, Thm. II), or Lojasiewicz (1965, §2, Thm. 2), it therefore satisfies local gradient domination in any stationary point. More precisely, for any stationary point $w_*$ there exist $\beta_{w_*} \in [\frac{1}{2}, 1]$ and $c_{w_*} > 0$ such that equation 4 in Definition 2.2 is satisfied. The local smoothness is inherently guaranteed by the fact that the loss function is analytic. To elaborate, every analytic function is infinitely differentiable, such that all derivatives remain locally bounded. Concerning Assumption 2.4, we assume that the underlying data in the supervised learning problem is compactly supported, ensuring that Assumption 2.4 is always satisfied locally.

In our notation, the stochastic first order oracle takes the form

$$V(w, (Z, Y)) = \nabla_w f^{\mathrm{DNN}}(w) + (\nabla_w \Phi(g_w(Z), Y) - \nabla_w f^{\mathrm{DNN}}(w)),$$

where we denote $\zeta = (Z, Y)$ and the iterative SGD then reads as

$$W_{n+1} = W_n - \gamma_n \nabla_w \Phi(g_{W_n}(Z_{n+1}), Y_{n+1})$$

with $\zeta_n = (Z_n, Y_n)$ independent and identical distributed. The iterative scheme of SHB can be written similarly. Note that this scenario also includes the empirical risk minimization of $\frac{1}{M} \sum_{m=1}^{M} \Phi(g_w(z^{(m)}), y^{(m)})$ when $\zeta = (Z, Y) \sim \frac{1}{M} \sum_{m=1}^{M} \delta_{(z^{(m)}, y^{(m)})}$, see Example 2.5 for more details. In this case, Assumption 2.4 is satisfied even with $A = B = 0$. Thus, the following local convergence is a direct consequence of Theorem 5.1.

**Corollary 6.1.** *Let $\delta > 0$. Denote by $(W_n)_{n \in \mathbb{N}}$ the sequence generated by SGD with $w \mapsto \nabla_w f^{\mathrm{DNN}}(w)$ as objective function, step size $\gamma_n \in \Theta(n^{-\theta})$ for $\theta \in (\frac{1}{2}, 1)$, and assume that $f^{\mathrm{DNN}}$ is analytic. Let $\mathcal{W}^*$ be an isolated compact set of local minima with level $l = f^{\mathrm{DNN}}(w^*)$ for all $w^* \in \mathcal{W}^*$ and suppose Assumption 2.4 is satisfied within $\mathcal{W}^*$. Suppose that $\gamma_n \leq \gamma_1$ for sufficiently small $\gamma_1$ (depending on $\delta$), then there exist two subsets $\mathcal{U}, \mathcal{U}_1$ of $\mathbb{R}^{d_w}$ such that $W_1 \in \mathcal{U}_1$ implies that the event $\Omega_{\mathcal{U}} = \{W_n \in \mathcal{U}, \text{ for all } n \geq 1\}$ has probability at least $1 - \delta$. Moreover, there exists $\beta \in [\frac{1}{2}, 1]$ such that for any*

$$\eta \in \begin{cases} \left( \max\{2 - 2\theta, \frac{\theta + 2\beta - 2}{2\beta - 1}\}, 1 \right) & : \beta \in (\frac{1}{2}, 1] \\ (2 - 2\theta, 1) & : \beta = \frac{1}{2} \end{cases}$$

*it holds that $|f^{\mathrm{DNN}}(W_n) - l| \mathbf{1}_\Omega \in o(n^{\eta - 1})$ almost surely and in expectation.*

In words: If the iterates of SGD reach a certain area around a local minimum, they are likely to become trapped in that region with high probability, provided that the step size is sufficiently small. This results shows that, under very general conditions, SGD converges to local minima and furthermore quantifies the convergence speed.

*Remark* 6.2. One may similarly apply Theorem 5.2 in the training of DNNs to derive convergence towards a global minimum with high probability provided that the initial loss $f^{\mathrm{DNN}}(X_1)$ and initial step size $\gamma_1$ are sufficiently small.

# 7 Application in Reinforcement Learning

Recent results showed that choosing the tabular softmax parametrization in policy gradient (PG) algorithms results in objective functions which fulfill a non-uniform gradient domination property (Mei et al., 2020; Yuan et al., 2022; Klein et al., 2024). While convergence of PG for exact gradients is well understood, convergence rates for stochastic PG are rare and mostly require very large batch sizes (Ding et al., 2022; Klein et al., 2024; Ding et al., 2023).

Let $(\mathcal{S}, \mathcal{A}, \rho, r, p)$ be a discounted MDP with finite state space $\mathcal{S}$, finite action space $\mathcal{A}$ and discount factor $\rho \in [0, 1)$. Further, $r : \mathcal{S} \times \mathcal{A} \to \mathbb{R}_+$ is the positive expected reward function and $p(s'|s, a)$ describes the transition probability from state $s$ to $s'$ under action $a$. As in Mei et al. (2020) we assume that the rewards are bounded in $[0, 1]$. Consider the stationary tabular softmax policy for parameter $w \in \mathbb{R}^{|\mathcal{S}||\mathcal{A}|}$, i.e.

$$\pi_w(a|s) = \frac{\exp(w(s, a))}{\sum_{a' \in \mathcal{A}_s} \exp(w(s, a'))}, \quad \forall s \in \mathcal{S}, a \in \mathcal{A}.$$

In the following we consider entropy regularized PG jointly with vanilla PG by setting $\lambda = 0$ when no regularization is considered. Then, for an initial state distribution $\mu$, the value function under the softmax parametrization is given by

$$V_\lambda^{\pi_w}(\mu) = \mathbb{E}_\mu^{\pi_w} \left[ \sum_{t=0}^{\infty} \rho^t r(S_t, A_t) \right] - \lambda \mathbb{E}_\mu^{\pi_w} \left[ \sum_{t=0}^{\infty} \rho^t \log(\pi_w(A_t | S_t)) \right]$$

and denote by $V_\lambda^*(\mu)$ the global optimum and by $\pi^*$ the optimal policy. If $\lambda > 0$ there exists a unique optimal policy which can be represented by the tabular softmax parametrization Nachum et al. (2017). Therefore, there must exist a continuum of optimal parameters $w^*$, such that $V_\lambda^{\pi_{w^*}}(\mu) = V_\lambda^*(\mu)$; see also the discussion in Ding et al. (2022). If $\lambda = 0$ no such parameters exists.

In order to maximize the objective we use stochastic gradient ascent. For $\lambda = 0$ this is called stochastic policy gradient method or REINFORCE. Note that e.g., Zhang et al. (2020) provide a stochastic first order oracle which meets the conditions required in Assumption 2.4. For $\lambda > 0$, a stochastic gradient estimator that satisfies Assumption 2.4 with $A = B = 0$ is presented in Ding et al. (2023, Eq. (4)). In both cases a non-uniform PL-inequality holds (Mei et al., 2020, Lem. 8, Lem. 15): For every $w \in \mathbb{R}^{|\mathcal{S}| \times |\mathcal{A}|}$ it holds that

$$\|\nabla_w V_\lambda^{\pi_w}(\mu)\|_2 \geq c_\lambda(w)^x \left[V_\lambda^*(\mu) - V_\lambda^{\pi_w}(\mu)\right]^x,$$

with $x = 1$ if $\lambda = 0$ and $x = \frac{1}{2}$ if $\lambda > 0$ and

$$c_\lambda(w) = \begin{cases} \frac{\min_{s \in \mathcal{S}} \pi_w(a^*(s)|s)}{\sqrt{|\mathcal{S}|}(1-\rho)} \left\|\frac{d_\mu^{\pi^*}}{\mu}\right\|_\infty^{-1}, & \lambda = 0, \\ \frac{2\lambda}{|\mathcal{S}|(1-\rho)} \min_s \mu(s) \min_{s,a} \pi_w(a|s)^2 \left\|\frac{d_\mu^{\pi^*}}{\mu}\right\|_\infty^{-1}, & \lambda > 0. \end{cases} \tag{16}$$

Here $a^*(s)$ denotes the best possible action in state $s$. W.l.o.g. we assume that $a^*(s)$ is unique, otherwise one can simply consider the maximum over all possible best actions, i.e. replace $\min_{s \in \mathcal{S}} \pi_w(a^*(s)|s)$ with $\min_{s \in \mathcal{S}} \min_{a \text{ is optimal action in } s} \pi_w(a|s)$. We prove that this implies a local gradient domination property with $\beta = 1$ (weak PL) for $\lambda = 0$ and $\beta = \frac{1}{2}$ (strong PL) for $\lambda > 0$.

**Proposition 7.1.** *There exists $r, c > 0$ such that for all $w \in \mathcal{B}_{r,\lambda}^* = \{w : V_\lambda^*(\mu) - V_\lambda^{\pi_w}(\mu) \leq r\}$ it holds that $c_\lambda(w) \geq c$.*

The proof of this proposition is given in Appendix E.

As the objective function $w \mapsto V_\lambda^{\pi_w}(\mu)$ is smooth and Lipschitz on $\mathbb{R}^{|\mathcal{S}||\mathcal{A}|}$ (see Yuan et al. (2022, Lem. E.1) for $\lambda = 0$ and Ding et al. (2023) for $\lambda > 0$), all assumptions in Theorem 5.2 are satisfied and we obtain the following result.

**Corollary 7.2.** *Let $\delta > 0$. Denote by $(W_n)_{n \in \mathbb{N}}$ the sequence generated by SGD with $w \mapsto -V_\lambda^{\pi_w}(\mu)$ as objective function, step size $\gamma_n \in \Theta(n^{-\theta})$ for $\theta \in (\frac{1}{2}, 1)$ and suppose $\gamma_n \leq \gamma_1$ for sufficiently small $\gamma_1$ (depending on $\delta$). Then, there exist two subsets $\mathcal{U}, \mathcal{U}_1$ of $\mathbb{R}^{|\mathcal{S}||\mathcal{A}|}$ such that $W_1 \in \mathcal{U}_1$ implies that the event $\Omega_\mathcal{U} = \{W_n \in \mathcal{U}, \text{ for all } n \geq 1\}$ has probability at least $1 - \delta$. Moreover, for any*

$$\eta \in \begin{cases} (\max\{2 - 2\theta, \theta\}, 1), & \text{if } \lambda = 0, \\ (2 - 2\theta, 1), & \text{if } \lambda > 0 \end{cases}$$

*it holds that $(V_\lambda^*(\mu) - V_\lambda^{\pi_{W_n}}(\mu))\mathbf{1}_\Omega \in o(n^{\eta-1})$ almost surely and in expectation.*

In words: If the (regularized) stochastic policy gradient algorithm is started close enough to the optimum a nearly $o(n^{-\frac{1}{3}})$ ($o(n^{-1})$ respectively) almost sure rate of convergence can be obtained by choosing $\theta = \frac{2}{3}$ ($\theta$ close to 1 respectively) This is in contrast to $o(n^{-1})$ (linear convergence) known in (regularized) policy gradient with access to exact gradients.

*Remark* 7.3. Note that $r$ and $c$ in Lemma 7.1 can be explicitly chosen (see Remark E.1). Hence, one can choose the neighbourhoods $\mathcal{U}$ and $\mathcal{U}_1$ w.r.t. $r$ as in Equation (39) and Lemma D.10 in Appendix D to find an explicit neighbourhood $\mathcal{U}_1$ as condition for initialization.

Our convergence result in Corollary 7.2 extends the local convergence analysis of stochastic policy gradient under entropy regularization from Ding et al. (2023). Like their work, our analysis builds on Mertikopoulos et al. (2020). However, while Ding et al. (2023) require an increasing batch size sequence to ensure convergence to the regularized optimum, we achieve local convergence for both the unregularized and entropy-regularized settings without this requirement. For $\lambda > 0$, our method attains convergence arbitrarily close to $o(\frac{1}{n})$ that matches the rate obtained by Ding et al. (2023) while avoiding the need to increase batch sizes. Finally, we highlight that our local convergence holds almost surely on an event with high probability.

**Acknowledgements**  Sara Klein thankfully acknowledges the funding support by the Hanns-Seidel-Stiftung e.V. and is grateful to the DFG RTG1953 "Statistical Modeling of Complex Systems and Processes" for funding this research. Waïss Azizian was supported in part by the French National Research Agency (ANR) in the framework of the PEPR IA FOUNDRY project (ANR-23-PEIA-0003) and MIAI@Grenoble Alpes (ANR-19-P3IA-0003).

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

# A   Auxiliary Convergence Theorems

In the following section, we provide two specific convergence theorems used to prove almost sure convergence (Lemma A.2) as well as convergence in expectation (Lemma A.3). The former one is a direct consequence of the well-known Robbins-Siegmund theorem, provided here for completeness.

**Theorem A.1** (Theorem 1 in Robbins and Siegmund (1971)). *Let $(\Omega, \mathcal{F}, (\mathcal{F}_n)_{n \in \mathbb{N}}, \mathbb{P})$ be a filtered probability space, $(Z_n)_{n \in \mathbb{N}}$, $(A_n)_{n \in \mathbb{N}}$, $(B_n)_{n \in \mathbb{N}}$ and $(C_n)_{n \in \mathbb{N}}$ be non-negative and adapted stochastic processes with*

$$\sum_{n=1}^{\infty} A_n < \infty \quad and \quad \sum_{n=1}^{\infty} B_n < \infty$$

*almost surely. Suppose that for each $n \in \mathbb{N}$ the recursion*

$$\mathbb{E}[Z_{n+1} \mid \mathcal{F}_n] \leq (1 + A_n)Z_n + B_n - C_n$$

*is satisfied, then (i) there exists an almost surely finite random variable $Z_\infty$ such that $Z_n \to Z_\infty$ almost surely as $n \to \infty$ and (ii) $\sum_{n=1}^{\infty} C_n < \infty$ almost surely.*

**Lemma A.2.** *Let $(\Omega, \mathcal{F}, (\mathcal{F}_n)_{n \in \mathbb{N}}, \mathbb{P})$ be a filtered probability space, $(Y_n)_{n \in \mathbb{N}}$, $(a_n)_{n \in \mathbb{N}}$, $(b_n)_{n \in \mathbb{N}}$ and $(r_n)_{n \in \mathbb{N}}$ be non-negative and adapted stochastic processes with*

$$\sum_{n=1}^{\infty} a_n = \infty, \quad \sum_{n=1}^{\infty} b_n < \infty \quad and \quad r_n > 0$$

*almost surely. Suppose that for each $n \in \mathbb{N}$ the recursion*

$$\mathbb{E}[r_{n+1}Y_{n+1} \mid \mathcal{F}_n] \leq (1 - a_n)r_nY_n + b_n$$

*is satisfied, then we have $r_nY_n \to 0$ almost surely as $n \to \infty$.*

*Proof.* We define $Z_n := r_nY_n$, $B_n := b_n$ and $C_n := a_nr_nY_n$ such that

$$\mathbb{E}[Z_{n+1} \mid \mathcal{F}_n] \leq Z_n - C_n + B_n$$

for $n \in \mathbb{N}$. Using Lemma A.1 we observe that there exists $Z_\infty$ almost surely finite such that $Z_n = r_nY_n \to Z_\infty$ almost surely as $n \to \infty$. Moreover, we obtain that

$$\sum_{n=1}^{\infty} C_n = \sum_{n=1}^{\infty} a_nr_nY_n < \infty$$

almost surely, which yields that

$$\liminf_{n \to \infty} r_nY_n = 0$$

almost surely, since $\sum_{n=1}^{\infty} a_n = \infty$ almost surely. Since limit inferior and limit coincide for converging sequences, the assertion follows:

$$Z_\infty = \lim_{n \to \infty} r_nY_n = \liminf_{n \to \infty} r_nY_n = 0$$

almost surely. $\qquad\qquad\square$

The following Lemma will be applied to prove convergence in expectation.

**Lemma A.3.** *Let $(w_n)_{n \in \mathbb{N}}$ be a non-negative sequence, such that $w_{n+1} \leq (1 - a_n)w_n + b_n$, where $(a_n)_{n \in \mathbb{N}}$ and $(b_n)_{n \in \mathbb{N}}$ are non-negative sequences satisfying*

$$\sum_{n=1}^{\infty} a_n = \infty \quad and \quad \sum_{n=1}^{\infty} b_n < \infty.$$

*Then, $\lim_{n \to \infty} w_n = 0$.*

*Proof.* W.l.o.g we assume that $w_{n+1} = (1 - a_n)w_n + b_n$, otherwise we could just increase $a_n$ or decrease $b_n$ which would have no effect on the summation tests. We obtain

$$-w_1 \leq w_n - w_1 = \sum_{k=1}^{n-1} (w_{k+1} - w_k) = \sum_{k=1}^{n-1} b_k - \sum_{k=1}^{n-1} w_k a_k.$$

Since $w_n - w_1$ is bounded below and $\sum_{k=1}^{\infty} b_k < \infty$, we deduce that $\sum_{k=1}^{n} w_k a_k$ is bounded. Since all summands are positive, the infinite sum converges. Thus, as a difference of two converging series also $(w_n)_{n \in \mathbb{N}}$ converges. Finally, the convergence of $\sum_{k=1}^{\infty} w_k a_k$ implies $\liminf_{n \to \infty} w_n = 0$ which, by the convergence of $(w_n)_{n \in \mathbb{N}}$, implies $\lim_n w_n = \liminf_n w_n = 0$. $\qquad\square$

# B   Numerical experiment - Toy example

In the following numerical experiment, we aim to verify our theoretical finding. We have implemented the same toy example similar to Fatkhullin et al. (2022) to test our theoretical findings. In our implementation, we consider both SGD and SHB applied to the objective function $f_p(x) = |x|^p$, where $x \in \mathbb{R}$, for various choices of $p \geq 2$. It is straightforward to verify that $f_p$ satisfies the global gradient domination with parameter $\beta(p) = \frac{p-1}{p}$. It is noteworthy that for $p = 2$, the $f_p$ obviously satisfies the PL condition with $\beta = \frac{1}{2}$, whereas for increasing $p \to \infty$, we move towards $\beta(p) \to 1$. We have used the step size schedule $\Theta(n^{-\frac{2\beta(p)}{4\beta(p)-1}})$ discussed in Table 1 and observed the almost sure convergence rates $n^{-\frac{1}{4\beta(p)-1}}$ as suggested by Theorem 4.1 and Theorem 4.2. Note that our derived rates are arbitrarily close to the sharp upper bound known in expectation Fatkhullin et al. (2022).

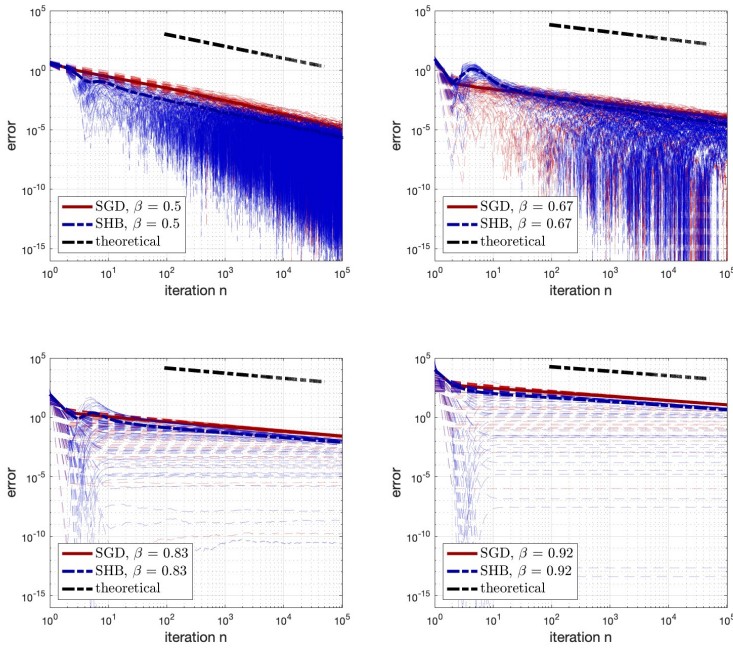

Figure 1: Pathwise error $(f_p(X_n))_{n=1,\dots,N}$ of SGD and SHB for various choices of $\beta \in \{0.5, 0.67, 0.83, 0.92\}$. For each setting we have simulated 100 runs of length , $N = 10^5$. The bold lines correspond to the average error of SGD (red) and SHB (blue), and the black dash-dotted line corresponds to the theoretical rate $n^{-\frac{1}{4\beta-1}}$.

**Details of the implementation:**   Both algorithms have been implemented by hand using `MATLAB`. We have initialized both SGD and SHB with the initial state $X_1 \sim \frac{1}{2}\mathcal{U}([1.5, 2.5]) + \frac{1}{2}\mathcal{U}([-2.5, 1.5])$ to force initials

which are not close to the actual minimum $x^* = 0$. The initial step sizes $\gamma_1(\beta)$ for both algorithms are chosen as

$$\gamma_1(0.5) = 0.2, \ \gamma_1(0.67) = 0.13, \ \gamma_1(0.83) = 0.004, \ \gamma_1(0.92) = 10^{-6}$$

through which we counteract the decreasing smoothness for $\beta \to 1$. The momentum parameter for SHB is fixed for all $\beta$ as $\nu = 0.5$. The exact gradients $\nabla f_p$ are perturbed by independent additive noise following a standard normal distribution $\mathcal{N}(0, 1)$.

## C   Proof of Lemma 3.1

In the following section, we present the proof of our super-martingale result.

*Proof of Lemma 3.1.* In the following, we treat both cases $\beta = \frac{1}{2}$ and $\beta \in (\frac{1}{2}, 1]$ separately.
$\underline{\beta = \frac{1}{2}}$: In this case, the inequality reduces to

$$\mathbb{E}[Y_{n+1} \mid \mathcal{F}_n] \leq (1 + c_1\gamma_n^2 - c_2\gamma_n)Y_n + c_3\gamma_n^2.$$

By the choice of $\gamma_n$, there exists some $N > 0$ and $\tilde{c}_1 > 0$ such that $c_2\gamma_n - c_1\gamma_n^2 \geq \tilde{c}_1\gamma_n$ for all $n \geq N$. Hence, for all $n \geq N$

$$\mathbb{E}[Y_{n+1} \mid \mathcal{F}_n] \leq (1 - \tilde{c}_1\gamma_n)Y_n + c_3\gamma_n^2$$

such that the claim follows by Liu and Yuan (2022, Lem. 1).

$\underline{\beta \in (\frac{1}{2}, 1]}$: The proof uses the elementary inequality

$$(n + 1)^{1-\eta} \leq n^{1-\eta} + (1 - \eta)n^{-\eta}, \tag{17}$$

which was also applied and proved in Liu and Yuan (2022, Lem. 1). The aim is to apply the Robbins-Siegmund implication, Lemma A.2, in order to derive the almost sure convergence rate. Let $1 \leq q < 2$ be arbitrary for now. The key step of the proof is the following computation

$$\begin{aligned}
\mathbb{E}[Y_{n+1} \mid \mathcal{F}_n] &\leq (1 + c_1\gamma_n^2)Y_n - c_2\gamma_n Y_n^{2\beta} + c_3\gamma_n^2 \\
&= (1 + c_1\gamma_n^2)Y_n - c_2\gamma_n^q Y_n + c_2\gamma_n^q Y_n - c_2\gamma_n Y_n^{2\beta} + c_3\gamma_n^2 \\
&= (1 + c_1\gamma_n^2 - c_2\gamma_n^q)Y_n + c_2\gamma_n\left(\gamma_n^{q-1}Y_n - Y_n^{2\beta}\right) + c_3\gamma_n^2.
\end{aligned} \tag{18}$$

Similar to the case $\beta = \frac{1}{2}$ there exists some $N > 0$ and $\tilde{c}_1 > 0$ such that $c_2\gamma_n^q - c_1\gamma_n^2 \geq \tilde{c}_1\gamma_n^q$ for all $n \geq N$. Hence, for all $n \geq N$ we obtain the iterative inequality of the form

$$\mathbb{E}[Y_{n+1} \mid \mathcal{F}_n] \leq (1 - \tilde{c}_1\gamma_n^q)Y_n + c_2\gamma_n\left(\gamma_n^{q-1}Y_n - Y_n^{2\beta}\right) + c_3\gamma_n^2. \tag{19}$$

The function $x \mapsto ax - bx^{2\beta}$ takes it maximum at $\bar{x} = \left(\frac{a}{2b\beta}\right)^{\frac{1}{2\beta-1}}$ such that

$$\begin{aligned}
\gamma_n(\gamma_n^{q-1}Y_n - Y_n^{2\beta}) &\leq \frac{\gamma_n^{q+\frac{q-1}{2\beta-1}}}{(2\beta)^{\frac{1}{2\beta-1}}} - \frac{\gamma_n^{1+\frac{(q-1)2\beta}{2\beta-1}}}{(2\beta)^{\frac{2\beta}{2\beta-1}}} \\
&= \frac{1}{(2\beta)^{\frac{1}{2\beta-1}}}\gamma_n^{\frac{2q\beta-1}{2\beta-1}} - \frac{1}{(2\beta)^{\frac{2\beta}{2\beta-1}}}\gamma_n^{\frac{2q\beta-1}{2\beta-1}} \\
&= (2\beta)^{-\frac{1}{2\beta-1}}(1 - \frac{1}{2\beta})\gamma_n^{\frac{2q\beta-1}{2\beta-1}}
\end{aligned} \tag{20}$$

holds almost surely. We define $\tilde{c}_2 = c_2(2\beta)^{-\frac{1}{2\beta-1}}(1 - \frac{1}{2\beta}) \in (0, \infty)$ for $\beta \in (\frac{1}{2}, 1)$ and proceed with

$$\mathbb{E}[Y_{n+1} \mid \mathcal{F}_n] \leq (1 - \tilde{c}_1\gamma_n^q)Y_n + \tilde{c}_2\gamma_n^{\frac{2\beta q-1}{2\beta-1}} + c_3\gamma_n^2. \tag{21}$$

Next, we apply the elementary inequality Equation (17) and choose $q$ such that $\frac{1}{2} < \theta \leq \frac{1}{q} \leq 1$. Moreover by the choice of $\gamma_n$, there exists some $c_4 > 0$ such that $\tilde{c}_1 \gamma_n^q \geq \frac{c_4}{t q \theta}$ for all $n \geq N$. It follows that for all $n \geq N$

$$\mathbb{E}[(n+1)^{1-\eta} Y_{n+1} \mid \mathcal{F}_n]$$

$$\leq (n+1)^{1-\eta}(1 - \tilde{c}_1 \gamma_n^q) Y_n + (n+1)^{1-\eta} \tilde{c}_2 \gamma_n^{\frac{2\beta q - 1}{2\beta - 1}} + (n+1)^{1-\eta} c_3 \gamma_n^2$$

$$\leq (n^{1-\eta} + (1-\eta) n^{-\eta})(1 - \frac{c_4}{n^{q\theta}}) Y_n + (n+1)^{1-\eta} \tilde{c}_2 \gamma_n^{\frac{2\beta q - 1}{2\beta - 1}} + (n+1)^{1-\eta} c_3 \gamma_n^2$$

$$= \left(1 + \frac{1-\eta}{n} - \frac{c_4}{n^{q\theta}} - \frac{c_4(1-\eta)}{n^{q\theta+1}}\right) n^{1-\eta} Y_n + (n+1)^{1-\eta} \tilde{c}_2 \gamma_n^{\frac{2\beta q - 1}{2\beta - 1}} + (n+1)^{1-\eta} c_3 \gamma_n^2.$$

We set $\tilde{c}_3 = \max\{\tilde{c}_2, c_3\}$ such that for all $n \geq N$

$$\mathbb{E}[(n+1)^{1-\eta} Y_{n+1} \mid \mathcal{F}_n]$$

$$\leq \left(1 + \frac{1-\eta}{n} - \frac{c_4}{n^{q\theta}} - \frac{c_4(1-\eta)}{n^{q\theta+1}}\right) n^{1-\eta} Y_n + \tilde{c}_3 (n+1)^{1-\eta}(\gamma_n^{\frac{2\beta q - 1}{2\beta - 1}} + \gamma_n^2).$$

Observe that $q\theta \leq 1$ by condition $\theta \leq \frac{1}{q}$. Hence, there exists $\tilde{c}_4 > 0$ and $\tilde{N} > N$ for sufficiently large $\tilde{N} \geq N$ such that for all $n \geq \tilde{N}$ we have

$$\mathbb{E}[(n+1)^{1-\eta} Y_{n+1} \mid \mathcal{F}_n] \leq (1 - \tilde{c}_4 \frac{1}{n^{q\theta}}) n^{1-\eta} Y_n + c_3 (n+1)^{1-\eta}(\gamma_n^{\frac{2\beta q - 1}{2\beta - 1}} + \gamma_n^2) \qquad (22)$$

In order to apply Robbins-Siegmund, more precisely Lemma A.2, we are going to verify the following three sufficient conditions:

$$\sum_{n=\tilde{N}}^{\infty} \frac{1}{n^{q\theta}} = \infty, \qquad (23)$$

$$\sum_{n=\tilde{N}}^{\infty} n^{1-\eta-2\theta} < \infty, \qquad (24)$$

$$\sum_{n=\tilde{N}}^{\infty} n^{1-\eta-\frac{\theta(2\beta q - 1)}{2\beta - 1}} < \infty. \qquad (25)$$

Then, $Y_n \in o\left(\frac{1}{n^{1-\eta}}\right)$ almost surely.

The first condition Equation (23) is obviously satisfied, since we assume $\theta \leq \frac{1}{q}$. For the second condition Equation (24) we may choose $\theta > 1 - \frac{\eta}{2}$ such that $1 - \eta - 2\theta < -1$. The third condition Equation (25) gives $1 - \eta - \frac{\theta(2\beta q - 1)}{2\beta - 1} < -1$ which leads to the condition $\theta > \frac{(2-\eta)(2\beta - 1)}{2\beta q - 1}$. Hence, all together we obtain the sufficient condition

$$\theta \in \left(\max\left\{\frac{(2-\eta)(2\beta - 1)}{2\beta q - 1}, 1 - \frac{\eta}{2}\right\}, \frac{1}{q}\right].$$

In the following, we consider the two cases separately that correspond to the maximum being either $1 - \frac{\eta}{2}$ or $\frac{(2-\eta)(2\beta - 1)}{2\beta q - 1}$. The first case occurs precisely for $\frac{1}{q} \leq \frac{2\beta}{4\beta - 1}$, the latter one for $\frac{1}{q} \geq \frac{2\beta}{4\beta - 1}$.

Firstly, let $\frac{1}{q} \leq \frac{2\beta}{4\beta - 1}$. In this situation the sufficient condition on $\theta$ simplifies to

$$\theta \in \left(1 - \frac{\eta}{2}, \frac{1}{q}\right].$$

The interval is non-empty for $\frac{1}{q} > \frac{2-\eta}{2}$, which requires $\eta \in (\frac{4\beta - 2}{4\beta - 1}, 1)$.

Secondly, let $\frac{1}{q} \geq \frac{2\beta}{4\beta - 1}$. In this situation the sufficient condition on $\theta$ simplifies to

$$\theta \in \left(\frac{(2-\eta)(2\beta - 1)}{2\beta q - 1}, \frac{1}{q}\right],$$

the interval is non-empty for $\frac{1}{q} < 2\beta\eta - 2\beta + 2 - \eta$. Hence, $\frac{1}{q} \in (\frac{2\beta}{4\beta-1}, 2\beta\eta - 2\beta + 2 - \eta)$ which requires the condition $\eta \in (\frac{4\beta-2}{4\beta-1}, 1)$.

Either case yields sufficient conditions on $\theta$ and $\eta$ (depending on the auxiliary variable $q$) under which $Y_n \in o\left(\frac{1}{n^{1-\eta}}\right)$ holds almost surely. We will now utilize the free variable $q$ to prove the claim.

- Let $\theta \in (\frac{1}{2}, \frac{2\beta}{2\beta-1})$: We set $q = \frac{4\beta-1}{2\beta}$ and use the first case. The assumption $\eta > 2 - 2\theta = \max\{2 - 2\theta, \frac{\theta+2\beta-2}{2\beta-1}\}$ implies $\theta \in \left(1 - \frac{\eta}{2}, \frac{1}{q}\right]$. (Note that $\eta > \frac{4\beta-2}{4\beta-1}$ is automatically fulfilled by $2 - 2\theta > \frac{4\beta-2}{4\beta-1}$ for this choice of $\theta$.)

- Let $\theta \in [\frac{2\beta}{4\beta-1}, 1)$: By assumption we have $\eta > \frac{\theta+2\beta-2}{2\beta-1} = \max\{2 - 2\theta, \frac{\theta+2\beta-2}{2\beta-1}\}$. We choose some $\frac{1}{q} \in (\theta, 2\beta\eta - 2\beta + 2 - \eta)$ and use the second case. (Note that $\eta > \frac{4\beta-2}{4\beta-1}$ again is automatically fulfilled by $\frac{\theta+2\beta-2}{2\beta-1} > \frac{4\beta-2}{4\beta-1}$ for this choice of $\theta$.)

All in all we have proved that $\theta \in (\frac{1}{2}, 2)$ implies $Y_n \in o\left(\frac{1}{n^{1-\eta}}\right)$ almost surely for all $\eta \in (\max\{2 - 2\theta, \frac{\theta+2\beta-2}{2\beta-1}\}, 1)$. $\qquad\square$

## D  Proofs of Section 5

In the following section, we present the proofs of Section 5.

### D.1  Proof of Theorem 5.1

Suppose that the assumptions of Theorem 5.1 hold throughout this section. In contrast to the global gradient domination analysis we may assume w.l.o.g. the uniform second moment bounds, i.e. $A = B = 0$, instead of the more general Equation (ABC) condition. Choosing $A, B > 0$ would imply the bounded variance assumption of the gradient estimator. Note therefore, that the first term $A(f(x) - f(x^*))$ and the second term $B\|\nabla f(x)\|^2$ are both locally bounded by the local Lipschitz assumptions on $f$ and $\nabla f$.

Note that every isolated local minimum $\{x^*\}$ is a special case of an isolated compact connected set of local minima. In this case it holds that $\beta = \beta_{x^*}$. If $\mathcal{X}^*$ contains more then one point, we can unify the gradient domination property in a neighbourhood of $\mathcal{X}^*$ due to compactness. The set $\mathcal{X}^*$ has to be connected to assure that all local minima are on the same level $l$.

Recall, the outline of the proof is structured as follows:

- First, we unify the gradient domination property around the set of local minima $\mathcal{X}^*$ and obtain a radius $r$ such that the unified gradient domination property is fulfilled in all open balls with radius $r$ around $x^* \in \mathcal{X}^*$ (Lemma D.1).

- Based on this we construct sets $\mathcal{U}, \mathcal{U}_1 \subseteq \mathbb{R}^d$ and the events $\Omega_n \in \Omega$ (see Equation (26), Equation (27) and Equation (28)), such that $\Omega_{\mathcal{U}} = \bigcap_n \Omega_n$ occurs with high probability. To be precise, $\mathcal{U}_1$ and $\mathcal{U}$ are neighborhoods of $\mathcal{X}^*$ constructed such that the gradient domination property holds within this region, and when starting in $\mathcal{U}_1$ the gradient trajectory does remain in $\mathcal{U}$ for all gradient steps with high probability. Then, $\Omega_n$ describes the event that $X_k \in \mathcal{U}$ for all $k \leq n$.

- All following Lemmata before the proof of Theorem 5.1 are devoted to show that $\mathbb{P}(\Omega_n) \geq 1 - \delta$ for all $n \in \mathbb{N}$. This then proves Claim (i) of the Theorem. Claim (ii) and (iii) will be shown directly in the proof of Theorem 5.1 at the end of this subsection.

- In order to show $\mathbb{P}(\Omega_n) \geq 1 - \delta$ we construct set $C_n$ and $E_n$ defined in Equation (29) and Equation (34) such that $E_n \cap C_n \subset \Omega_{n+1}$ (Lemma D.6) while Lemma D.5 is used to prove this claim.

- The sets $E_n$ are such that $f(X_n)$ remains close to $f^*$. We exploit the unified gradient domination property to construct the sets $E_n$ (Lemma D.4) and derive a recursive inequality in Lemma D.6 c) to prove that this event occurs with high probability (Lemma D.7).

- The sets $C_n$ are such that $X_{n+1}$ remains close to $X_n$ and we exploit the finite variance assumption to show that these events occur with high probability (Lemma D.8).

We denote by

$$\tilde{\mathcal{B}}_r(x) = \{y \in \mathbb{R}^d \,:\, ||x - y|| < r\}$$

the open ball with radius $r > 0$ around $x \in \mathbb{R}^d$ and by

$$\mathcal{B}_r(x) = \{y \in \mathbb{R}^d \,:\, ||x - y|| \leq r\}$$

the closed ball with radius $r > 0$ around $x \in \mathbb{R}^d$.

In the following Lemma we unify the gradient domination property around the set of local minima $\mathcal{X}^* \subset \mathbb{R}^d$.

**Lemma D.1.** . *There exists $r > 0$, $\beta \in [\frac{1}{2}, 1]$ and $c > 0$, such that for all $x \in \bigcup_{x^* \in \mathcal{X}^*} \tilde{\mathcal{B}}_r(x^*)$ it holds that*

$$f(x) > l \ \ for \ \ x \notin \mathcal{X}^* \quad and \quad ||\nabla f(x)|| \geq c(f(x) - l)^{\beta}\,.$$

*Proof.* By the local gradient domination property, for every $x^* \in \mathcal{X}^*$ there exist $r_{x^*} > 0$, $\beta_{x^*} \in [\frac{1}{2}, 1]$ and $c_{x^*} > 0$ such that

$$||\nabla f(x)|| \geq c_{x^*} |f(x) - l|^{\beta_{x^*}}, \quad \forall x \in \mathcal{B}_{r_{x^*}}(x^*).$$

Moreover, w.l.o.g we can assume that $f(x) > l$ for all $x \in \mathcal{B}_{r_{x^*}}(x^*) \setminus \mathcal{X}^*$, as $\mathcal{X}^*$ is an isolated compact connected set of local minima (otherwise choose $r_{x^*}$ small enough).

By the compactness of $\mathcal{X}^*$ we can find a finite subset $\mathcal{Y}^* \subset \mathcal{X}^*$, such that

$$\tilde{\mathcal{U}} := \bigcup_{y^* \in \mathcal{Y}^*} \tilde{\mathcal{B}}_{r_{y^*}}(y^*) \supset \mathcal{X}^*.$$

Then, we define $\beta = \max_{y^* \in \mathcal{Y}^*} \beta_{y^*}$ and $c = \min_{y^* \in \mathcal{Y}^*} c_{y^*}$. For any $x \in \tilde{\mathcal{U}}$ there exits $y^* \in \mathcal{Y}^*$ such that

$$||\nabla f(x)|| \geq c_{y^*}(f(x) - l)^{\beta_{y^*}} \geq c(f(x) - l)^{\beta}.$$

Thus, there exists an open neighbourhood $\tilde{\mathcal{U}}$ of $\mathcal{X}^*$ and $\beta \in [\frac{1}{2}, 1]$, $c > 0$, such that for all $x \in \tilde{\mathcal{U}}$ it holds that

$$f(x) > l \text{ for } x \notin \mathcal{X}^* \quad and \quad ||\nabla f(x)|| \geq c(f(x) - l)^{\beta}.$$

As $\tilde{\mathcal{U}}$ is open by definition and $\mathcal{X}^* \subset \tilde{\mathcal{U}}$, we can find a radius $r > 0$, such that $\bigcup_{x^* \in \mathcal{X}^*} \tilde{\mathcal{B}}_r(x^*) \subseteq \tilde{\mathcal{U}}$. This proves the claim. $\square$

*Remark* D.2. It is noteworthy that the unified gradient domination property obtained in the previous Lemma does not require an absolute value, as $f(x) \geq l$ for all $x \in \bigcup_{x^* \in \mathcal{X}^*} \tilde{\mathcal{B}}_r(x^*)$. This is crucial to obtain the recursive inequalities in Lemma D.4 and we will exploit this also in the proof of Theorem 5.1 to obtain the convergence rates.

In the following let $\mathbf{r} > 0$, $c > 0$ and $\beta \in [\frac{1}{2}, 1]$ chosen as in the previous Lemma, such that the unified gradient domination property holds for all $x \in \bigcup_{x^* \in \mathcal{X}^*} \tilde{\mathcal{B}}_{\mathbf{r}}(x^*)$. Further define

$$s = \inf \left\{ f(x) - l \,:\, x \in \bigcup_{x^* \in \mathcal{X}^*} \mathcal{B}_{\frac{3r}{4}}(x^*) \setminus \bigcup_{x^* \in \mathcal{X}^*} \tilde{\mathcal{B}}_{\frac{r}{2}}(x^*) \right\}.$$

**Lemma D.3.** *It holds that $s > 0$.*

*Proof.* If $s = 0$, then there exists a sequence $(x_n) \in \bigcup_{x^* \in \mathcal{X}^*} \mathcal{B}_{\frac{3\mathbf{r}}{4}}(x^*) \setminus \bigcup_{x^* \in \mathcal{X}^*} \tilde{\mathcal{B}}_{\frac{\mathbf{r}}{2}}(x^*)$ with $f(x_n) \to l$ for $n \to \infty$. By definition of the set and compactness (boundedness) of $\mathcal{X}^*$, the sequence $x_n$ is bounded:

$$||x_n|| \leq \frac{3\mathbf{r}}{4} + \sup_{x^* \in \mathcal{X}^*} ||x^*|| < \infty.$$

Hence, there is a convergent sub-sequence $(x_{n_k})$ with $x_{n_k} \to x$ for $k \to \infty$ and by continuity of $f$ it holds that $f(x) = l$. Further, it holds for all $x^* \in \mathcal{X}^*$ that $||x_n - x^*|| \geq \frac{\mathbf{r}}{2}$ for all $n \in \mathbb{N}$ such that $\inf_{x^* \in \mathcal{X}^*} ||x - x^*|| \geq \frac{\mathbf{r}}{2}$.

On the other hand, by construction we have that $x \in \overline{\bigcup_{x^* \in \mathcal{X}^*} \mathcal{B}_{\frac{3\mathbf{r}}{4}}(x^*) \setminus \bigcup_{x^* \in \mathcal{X}^*} \tilde{\mathcal{B}}_{\frac{\mathbf{r}}{2}}(x^*)} \subset \overline{\bigcup_{x^* \in \mathcal{X}^*} \mathcal{B}_{\frac{3\mathbf{r}}{4}}(x^*)} \subset \bigcup_{x^* \in \mathcal{X}^*} \tilde{\mathcal{B}}_{\mathbf{r}}(x^*)$. And as $f(y) > l$ for all $y \in \tilde{\mathcal{B}}_{\mathbf{r}}(x^*) \setminus \mathcal{X}^*$ we deduce from $f(x) = l$ that $x \in \mathcal{X}^*$. This is a contradiction to $\inf_{x^* \in \mathcal{X}^*} ||x - x^*|| \geq \frac{\mathbf{r}}{2}$. $\square$

We choose $\epsilon > 0$, such that $2\epsilon + \sqrt{\epsilon} < s$. We define the sets

$$\mathcal{U}_1 = \{x \in \mathbb{R}^d : \inf_{x^* \in \mathcal{X}^*} ||x - x^*|| < \frac{\mathbf{r}}{2}, f(x) - l \leq \frac{\epsilon}{2}\} \tag{26}$$

$$\mathcal{U} = \{x \in \mathbb{R}^d : \inf_{x^* \in \mathcal{X}^*} ||x - x^*|| < \frac{\mathbf{r}}{2}\} \tag{27}$$

which are subsets of $\mathbb{R}^d$ and the decreasing sequence of events

$$\Omega_n = \{X_k \in \mathcal{U} \text{ for all } k \leq n\} \tag{28}$$

$$C_n = \{||X_{k+1} - X_k|| \leq \frac{\mathbf{r}}{4} \text{ for all } k \leq n\}, \tag{29}$$

and $C_0 = \Omega$, which are measurable sets in $(\Omega, \mathcal{F}, \mathbb{P})$.

In order to prove Theorem 5.1 we will show that $\Omega_n$ has probability at least $1 - \delta$ for all $n \in \mathbb{N}$. To do this, we construct another sequence of events $(\hat{E}_n)$ with $\hat{E}_n \subset \Omega_n$ which occur with probability at least $1 - \delta$ for any $n \in \mathbb{N}$.

Therefore, we fix the notation $D_n := f(X_n) - l$, $\xi_{n+1} := -\langle \nabla f(X_n), Z(X_n, \zeta_{n+1}) \rangle$ and $\mathbf{1}_\mathcal{A}$ denotes the indicator function for a measurable set $\mathcal{A}$ in $(\Omega, \mathcal{F}, \mathbb{P})$, i.e. $\mathbf{1}_\mathcal{A}(\omega) = 1$ if $\omega \in \mathcal{A}$ and $\mathbf{1}_\mathcal{A}(\omega) = 0$ if $\omega \notin \mathcal{A}$. We prove the following (recursive) inequalities.

**Lemma D.4.** *If $\beta = \frac{1}{2}$, then it holds that*

$$D_{n+1}\mathbf{1}_{\Omega_n} \leq (1 - \gamma_n c^2)D_n\mathbf{1}_{\Omega_n} + \gamma_n \xi_{n+1}\mathbf{1}_{\Omega_n} + \frac{L\gamma_n^2}{2}\mathbf{1}_{\Omega_n}||V_{n+1}(X_n)||^2,$$

$$\leq D_1 \prod_{k=1}^{n}(1 - \gamma_k c^2)\mathbf{1}_{\Omega_n} + \sum_{k=1}^{n}\left(\prod_{j=k}^{n}(1 - \gamma_j c^2)\right)\gamma_k \xi_{k+1}\mathbf{1}_{\Omega_n} \tag{30}$$

$$+ \frac{L}{2}\sum_{k=1}^{n}\gamma_k^2||V_{k+1}(X_k)||^2\mathbf{1}_{\Omega_n}.$$

*If $\beta \in (\frac{1}{2}, 1]$, for any $1 \leq q < 2$, it holds that*

$$D_{n+1}\mathbf{1}_{\Omega_n} \leq (1 - \gamma_n^q c^2)D_n\mathbf{1}_{\Omega_n} + (2\beta)^{-\frac{1}{2\beta-1}}(1 - \frac{1}{2\beta})c^2\gamma_n^{\frac{2\beta q-1}{2\beta-1}} + \gamma_n \xi_{n+1}\mathbf{1}_{\Omega_n} + \frac{L\gamma_n^2}{2}||V_{n+1}(X_n)||^2\mathbf{1}_{\Omega_n}$$

$$\leq D_1 \prod_{k=1}^{n}(1 - \gamma_k^q c^2) + \tilde{c}\sum_{k=1}^{n}\gamma_k^{\frac{2\beta q-1}{2\beta-1}} + \sum_{k=1}^{n}\left(\prod_{j=k}^{n}(1 - \gamma_j^q c^2)\right)\gamma_k \xi_{k+1}\mathbf{1}_{\Omega_n} \tag{31}$$

$$+ \frac{L}{2}\sum_{k=1}^{n}\gamma_k^2||V_{k+1}(X_k)||^2\mathbf{1}_{\Omega_n},$$

*for $\tilde{c} = (2\beta)^{-\frac{1}{2\beta-1}}(1 - \frac{1}{2\beta})c^2$.*

*Proof.* From $L$-smoothness we can deduce that

$$D_{n+1} \leq D_n - \gamma_n \langle \nabla f(X_n), V_{n+1}(X_n) \rangle + \frac{L\gamma_n^2}{2} \|V_{n+1}(X_n)\|^2$$

$$= D_n - \gamma_n \|\nabla f(X_n)\|^2 - \gamma_n \langle \nabla f(X_n), Z(X_n, \zeta_{n+1}) \rangle + \frac{L\gamma_n^2}{2} \|V_{n+1}(X_n)\|^2$$

$$= D_n - \gamma_n \|\nabla f(X_n)\|^2 + \gamma_n \xi_{n+1} + \frac{L\gamma_n^2}{2} \|V_{n+1}(X_n)\|^2$$

for $Z(X_n, \zeta_{n+1})$ from Assumption 2.4 and $\xi_{n+1} = -\langle \nabla f(X_n), Z(X_n, \zeta_{n+1}) \rangle$.

We separate the two cases of $\beta$:

$\underline{\beta = \frac{1}{2}}$: Iterating this inequality and using $\mathbf{1}_{\Omega_{n+1}} \leq \mathbf{1}_{\Omega_n}$ it follows that

$$
\begin{aligned}
D_{n+1}\mathbf{1}_{\Omega_n} &\leq D_n \mathbf{1}_{\Omega_n} - \gamma_n \mathbf{1}_{\Omega_n} \|\nabla f(X_n)\|^2 + \gamma_n \mathbf{1}_{\Omega_n} \xi_{n+1} + \frac{L\gamma_n^2}{2} \mathbf{1}_{\Omega_n} \|V_{n+1}(X_n)\|^2 \\
&\leq D_n \mathbf{1}_{\Omega_n} - \gamma_n c^2 (f(X_n) - l)\mathbf{1}_{\Omega_n} + \gamma_n \mathbf{1}_{\Omega_n} \xi_{n+1} + \frac{L\gamma_n^2}{2} \mathbf{1}_{\Omega_n} \|V_{n+1}(X_n)\|^2 \\
&= (1 - \gamma_n c^2) D_n \mathbf{1}_{\Omega_n} + \gamma_n \xi_{n+1} \mathbf{1}_{\Omega_n} + \frac{L\gamma_n^2}{2} \mathbf{1}_{\Omega_n} \|V_{n+1}(X_n)\|^2, \\
&\leq D_1 \prod_{k=1}^{n} (1 - \gamma_k c^2) + \sum_{k=1}^{n} \left( \prod_{j=k}^{n} (1 - \gamma_j c^2) \right) \gamma_k \xi_{k+1} \mathbf{1}_{\Omega_n} \\
&\quad + \frac{L}{2} \sum_{k=1}^{n} \left( \prod_{j=k}^{n} (1 - \gamma_j c^2) \right) \gamma_k^2 \|V_{k+1}(X_k)\|^2 \mathbf{1}_{\Omega_n} \\
&\leq D_1 \prod_{k=1}^{n} (1 - \gamma_k c^2) + \sum_{k=1}^{n} \left( \prod_{j=k}^{n} (1 - \gamma_j c^2) \right) \gamma_k \xi_{k+1} \mathbf{1}_{\Omega_n} \\
&\quad + \frac{L}{2} \sum_{k=1}^{n} \gamma_k^2 \|V_{k+1}(X_k)\|^2 \mathbf{1}_{\Omega_n},
\end{aligned}
\tag{32}
$$

where we used that the unified gradient domination property holds for all $X_k$, $k \leq n$ on the event $\Omega_n$.

$\underline{\beta \in (\frac{1}{2}, 1]}$: Similarly, the unified gradient domination property yields the claimed inequality for any $1 \leq q < 2$:

$$\begin{aligned}
D_{n+1}\mathbf{1}_{\Omega_n} &\leq D_n\mathbf{1}_{\Omega_n} - \gamma_n\mathbf{1}_{\Omega_n}\|\nabla f(X_n)\|^2 + \gamma_n\mathbf{1}_{\Omega_n}\xi_{n+1} + \frac{L\gamma_n^2}{2}\mathbf{1}_{\Omega_n}\|V_{n+1}(X_n)\|^2 \\
&\leq D_n\mathbf{1}_{\Omega_n} - \gamma_n c^2(f(X_n)-l)^{2\beta}\mathbf{1}_{\Omega_n} + \gamma_n\mathbf{1}_{\Omega_n}\xi_{n+1} + \frac{L\gamma_n^2}{2}\mathbf{1}_{\Omega_n}\|V_{n+1}(X_n)\|^2 \\
&= D_n\mathbf{1}_{\Omega_n} - \gamma_n c^2 D_n^{2\beta}\mathbf{1}_{\Omega_n} + \gamma_n\xi_{n+1}\mathbf{1}_{\Omega_n} + \frac{L\gamma_n^2}{2}\mathbf{1}_{\Omega_n}\|V_{n+1}(X_n)\|^2, \\
&= (1 - \gamma_n^q c^2)D_n\mathbf{1}_{\Omega_n} + \gamma_n c^2(\gamma_n^{1-q}D_n - D_n^{2\beta})\mathbf{1}_{\Omega_n} + \gamma_n\xi_{n+1}\mathbf{1}_{\Omega_n} + \frac{L\gamma_n^2}{2}\|V_{n+1}(X_n)\|^2\mathbf{1}_{\Omega_n} \\
&\leq (1 - \gamma_n^q c^2)D_n\mathbf{1}_{\Omega_n} + (2\beta)^{-\frac{1}{2\beta-1}}(1 - \frac{1}{2\beta})c^2\gamma_n^{\frac{2\beta q-1}{2\beta-1}} + \gamma_n\xi_{n+1}\mathbf{1}_{\Omega_n} + \frac{L\gamma_n^2}{2}\|V_{n+1}(X_n)\|^2\mathbf{1}_{\Omega_n} \\
&\leq D_1\prod_{k=1}^n(1 - \gamma_k^q c^2)\mathbf{1}_{\Omega_n} + \tilde{c}\sum_{k=1}^n\left(\prod_{j=k}^n(1 - \gamma_j^q c^2)\right)\gamma_k^{\frac{2\beta q-1}{2\beta-1}} \\
&\quad + \sum_{k=1}^n\left(\prod_{j=k}^n(1 - \gamma_j^q c^2)\right)\gamma_k\xi_{k+1}\mathbf{1}_{\Omega_n} + \frac{L}{2}\sum_{k=1}^n\left(\prod_{j=k}^n(1 - \gamma_j^q c^2)\right)\gamma_k^2\|V_{k+1}(X_k)\|^2\mathbf{1}_{\Omega_n} \\
&\leq D_1\prod_{k=1}^n(1 - \gamma_k^q c^2)\mathbf{1}_{\Omega_n} + \tilde{c}\sum_{k=1}^n\gamma_k^{\frac{2\beta q-1}{2\beta-1}} + \sum_{k=1}^n\left(\prod_{j=k}^n(1 - \gamma_j^q c^2)\right)\gamma_k\xi_{k+1}\mathbf{1}_{\Omega_n} \\
&\quad + \frac{L}{2}\sum_{k=1}^n\gamma_k^2\|V_{k+1}(X_k)\|^2\mathbf{1}_{\Omega_n},
\end{aligned}$$
(33)

for $\tilde{c} = (2\beta)^{-\frac{1}{2\beta-1}}(1 - \frac{1}{2\beta})c^2$ from the function trick Equation (20) which we applied in the forth inequality. We also used that the unified gradient domination property holds for all $X_k$, $k \leq n$ on the event $\Omega_n$. $\qquad\square$

For $\beta \in (\frac{1}{2}, 1]$ we know from the proof of Lemma 3.1 that we can choose the auxiliary parameter $q$ from the previous lemma in such a way, that $\sum_{n=1}^\infty n^{1-\eta}\gamma_n^{\frac{2\beta q-1}{2\beta-1}}$ is convergent for all $\eta \in (\max\{2-2\theta, \frac{\theta+2\beta-2}{2\beta-1}\}, 1)$ (Condition (iii) to apply Lemma A.2). As $\eta < 1$, it follows that $\sum_{n=1}^\infty \gamma_n^{\frac{2\beta q-1}{2\beta-1}} < \infty$ holds true for all these choices of $q$. Now define

$$M_n = \sum_{k=1}^n\left(\prod_{j=k}^n(1 - \gamma_j c^2)\right)\gamma_k\xi_{k+1}\mathbf{1}_{\Omega_k}, \quad M_n^{(q)} = \sum_{k=1}^n\left(\prod_{j=k}^n(1 - \gamma_j^q c^2)\right)\gamma_k\xi_{k+1}\mathbf{1}_{\Omega_k}$$

$$\text{and} \quad S_n = \frac{L}{2}\sum_{k=1}^n\gamma_k^2\|V_{k+1}(X_k)\|^2\mathbf{1}_{\Omega_k}.$$

Then, $(M_n)$ and $(M_n^{(q)})$ are $(\mathcal{F}_{n+1})$-martingales with zero mean and $(S_n)$ is a $(\mathcal{F}_{n+1})$-sub-martingale by Assumption 2.4. Note that by the choice of $\gamma_n$ we have that $\sum_n\gamma_n^2 < \infty$ and hence $\mathbb{E}[S_n] < \infty$ for all $n \in \mathbb{N}$.

Next, define $R_n = M_n^2 + S_n$ and $R_n = (M_n^{(q)})^2 + S_n$ respectively (with some abuse of notation), for every $n \in \mathbb{N}$. Moreover, let

$$E_n = \{R_k < \epsilon \text{ for all } k \leq n\}.$$
(34)

which is an $\mathcal{F}_{n+1}$-measurable event on $(\Omega, \mathcal{F}, \mathbb{P})$. We define $R_0 = 0$ such that $E_0 = \Omega$.

Now let $\hat{E}_n = E_n \cap C_n$, then we will first show, that $\hat{E}_n$ fulfills the property $\hat{E}_n \subset \Omega_{n+1}$ for all $n \in \mathbb{N}$ in Lemma D.6 and then that $\hat{E}_n$ occurs with probability at least $1 - \delta$ in Lemma D.8.

To prove that $\hat{E}_n \subset \Omega_{n+1}$ we need one more auxiliary result.

**Lemma D.5.** *Suppose $x, y \in \mathbb{R}^d$ such that*

1. $\inf_{x^* \in \mathcal{X}^*} ||x - x^*|| < \frac{\mathbf{r}}{2}$,

2. $f(y) - l < s$,

3. $||x - y|| \le \frac{\mathbf{r}}{4}$.

*Then it holds that* $\inf_{x^* \in \mathcal{X}^*} ||y - x^*|| < \frac{\mathbf{r}}{2}$.

*Proof.* By triangle inequality we have that $\inf_{x^* \in \mathcal{X}^*} ||y - x^*|| \le \frac{3\mathbf{r}}{4}$, i.e there exists $x^* \in \mathcal{X}^*$ such that $||y - x^*|| \le \frac{3\mathbf{r}}{4}$. Suppose now, that $\inf_{x^* \in \mathcal{X}^*} ||y - x^*|| \ge \frac{\mathbf{r}}{2}$, this means that $y \in \bigcup_{x^* \in \mathcal{X}^*} \mathcal{B}_{\frac{3\mathbf{r}}{4}}(x^*) \setminus \bigcup_{x^* \in \mathcal{X}^*} \tilde{\mathcal{B}}_{\frac{\mathbf{r}}{2}}(x^*)$. By the definition of

$$s = \inf \left\{ f(z) - l \ : \ z \in \bigcup_{x^* \in \mathcal{X}^*} \mathcal{B}_{\frac{3\mathbf{r}}{4}}(x^*) \setminus \bigcup_{x^* \in \mathcal{X}^*} \tilde{\mathcal{B}}_{\frac{\mathbf{r}}{2}}(x^*) \right\}$$

this contradicts the second assumption $f(y) - l < s$. $\qquad\square$

We deduce the following relations on the constructed sets:

**Lemma D.6.** *For* $\beta \in (\frac{1}{2}, 1]$ *let* $\gamma_n \le \gamma_1$ *be sufficiently small such that* $\sum_{n=1}^{\infty} \gamma_n^{\frac{2\beta q - 1}{2\beta - 1}} < \frac{\epsilon}{2\tilde{c}}$, *and for* $\beta = \frac{1}{2}$ *let* $\gamma_1 > 0$ *be arbitrary. Furthermore, assume that the initial* $X_1 \in \mathcal{U}_1$ *almost surely. Then,*

a) $E_{n+1} \subset E_n$, $\hat{E}_{n+1} \subset \hat{E}_n$ *and* $\Omega_{n+1} \subset \Omega_n$

b) $\hat{E}_n \subset \Omega_{n+1}$

c) *Define the events* $\tilde{E}_n = E_{n-1} \setminus E_n = E_{n-1} \cup \{R_n \ge \epsilon\}$. *Then, for* $\tilde{R}_n = R_n \mathbf{1}_{E_{n-1}}$, *there exists a* $\tilde{C} > 0$ *such that*

$$\mathbb{E}[\tilde{R}_n] \le \mathbb{E}[\tilde{R}_{n-1}] + \gamma_n^2 [G^2 C^2 + G^2 + C] - \epsilon \mathbb{P}(\tilde{E}_{n-1}).$$

*Proof.* a) Follows by definition of the events.

b) Note that $\hat{E}_0 = \Omega = \Omega_1$ because

$$X_1 \in \mathcal{U}_1 = \{x : \inf_{x^* \in \mathcal{X}^*} ||x - x^*|| < \frac{\mathbf{r}}{2}, f(x) - l \le \frac{\epsilon}{2}\} \subset \{x : \inf_{x^* \in \mathcal{X}^*} ||x - x^*|| < \frac{\mathbf{r}}{2}\} = \Omega_1$$

almost surely. We prove the assertion by induction. Let $\omega \in \hat{E}_n$. Since $\hat{E}_n \subset \hat{E}_{n-1} \subset \Omega_n$ by induction assumption, we have $\omega \in \Omega_n$ and thus $\omega \in \Omega_k$ for all $k \le n$. We will apply Lemma D.5 with $x = X_n(\omega)$ and $y = X_{n+1}(\omega)$. By definition it holds that $\omega \in \hat{E}_n$ implies condition 3. and $\omega \in \Omega_n$ implies condition 1. of Lemma D.5. It remains to show condition 2., then it follows that $\inf_{x^* \in \mathcal{X}^*} ||X_{n+1}(\omega) - x^*|| < \frac{\mathbf{r}}{2}$, i.e. $X_{n+1}(\omega) \in \mathcal{U}$ and by $\omega \in \Omega_n$ we deduce $\omega \in \Omega_{n+1}$.

To Prove condition 2. we separate both cases for $\beta$:

$\underline{\beta = \frac{1}{2}}$: The inequality Equation (30) and the induction hypothesis yield

$$D_{n+1}(\omega) = D_{n+1}(\omega)\mathbf{1}_{\Omega_n}(\omega)$$

$$\leq D_1(\omega)\prod_{k=1}^n(1-\gamma_k c^2) + \sum_{k=1}^n\left(\prod_{j=k}^n(1-\gamma_j c^2)\right)\gamma_k\xi_{k+1}(\omega)\mathbf{1}_{\Omega_n}(\omega)$$

$$+ \frac{L}{2}\sum_{k=1}^n\gamma_k^2\|V_{k+1}(X_k(\omega))\|^2\mathbf{1}_{\Omega_n}(\omega)$$

$$= D_1(\omega)\prod_{k=1}^n(1-\gamma_k c^2) + \sum_{k=1}^n\left(\prod_{j=k}^n(1-\gamma_j c^2)\right)\gamma_k\xi_{k+1}(\omega)\mathbf{1}_{\Omega_k}(\omega)$$

$$+ \frac{L}{2}\sum_{k=1}^n\gamma_k^2\|V_{k+1}(X_k(\omega))\|^2\mathbf{1}_{\Omega_k}(\omega)$$

$$\leq \frac{\epsilon}{2} + \sqrt{R_n(\omega)} + R_n(\omega)$$

$$\leq 2\epsilon + \sqrt{\epsilon} < s,$$

where the equation in the third line is due to $\omega \in \Omega_k$ for all $k \leq n$ by induction.

$\underline{\beta \in (\frac{1}{2}, 1]}$: Similarly, we obtain from Equation (42)

$$D_{n+1}(\omega) = D_{n+1}(\omega)\mathbf{1}_{\Omega_n}(\omega)$$

$$\leq D_1(\omega)\mathbf{1}_{\Omega_n}(\omega)\prod_{k=1}^n(1-\gamma_k^q c^2) + \tilde{c}\sum_{k=1}^n\gamma_k^{\frac{2\beta q-1}{2\beta-1}} + \sum_{k=1}^n\left(\prod_{j=k}^n(1-\gamma_j^q c^2)\right)\gamma_k\xi_{k+1}(\omega)\mathbf{1}_{\Omega_n}(\omega)$$

$$+ \frac{L}{2}\sum_{k=1}^n\gamma_k^2\|V_{k+1}(X_k(\omega))\|^2\mathbf{1}_{\Omega_n}(\omega)$$

$$= D_1(\omega)\prod_{k=1}^n(1-\gamma_k^q c) + \sum_{k=1}^n\left(\prod_{j=k}^n(1-\gamma_j c^2)\right)\gamma_k\xi_{k+1}(\omega)\mathbf{1}_{\Omega_k}(\omega)$$

$$+ \frac{L}{2}\sum_{k=1}^n\gamma_k^2\|V_{k+1}(X_k(\omega))\|^2\mathbf{1}_{\Omega_k}(\omega)$$

$$\leq \frac{\epsilon}{2} + \frac{\epsilon}{2} + \sqrt{R_n(\omega)} + R_n(\omega)$$

$$\leq 2\epsilon + \sqrt{\epsilon} < s.$$

We used in both cases that that $\prod_{k=1}^n(1-\gamma_k^q c) \leq 1$ and the choice of $\epsilon$ such that $2\epsilon + \sqrt{\epsilon} < s$. This proves that condition 2. in Lemma D.5 is also satisfied which concludes the induction.

c) Without loss of generality we consider the case $\beta = 1/2$. The computations for $\beta \in (1/2, 1]$ follow in line by replacing $M_n$ with $M_n^{(q)}$. By definition it holds that $E_n = E_{n-1} \setminus (E_{n-1} \setminus E_n) = E_{n-1} \setminus \tilde{E}_n$. Then we have

$$\tilde{R}_n = R_n\mathbf{1}_{E_{n-1}}$$
$$= R_{n-1}\mathbf{1}_{E_{n-1}} + (R_n - R_{n-1})\mathbf{1}_{E_{n-1}}$$
$$= R_{n-1}\mathbf{1}_{E_{n-2}} - R_{n-1}\mathbf{1}_{\tilde{E}_{n-1}} + (R_n - R_{n-1})\mathbf{1}_{E_{n-1}}$$
$$= \tilde{R}_{n-1} - R_{n-1}\mathbf{1}_{\tilde{E}_{n-1}} + (R_n - R_{n-1})\mathbf{1}_{E_{n-1}}$$

and for the last term

$$R_n - R_{n-1} = M_n^2 - M_{n-1}^2 + S_n - S_{n-1}$$

$$= \gamma_n^2(1-\gamma_n c^2)^2\xi_{n+1}^2\mathbf{1}_{\Omega_n} + 2\gamma_n(1-\gamma_n c)\xi_{n+1}\mathbf{1}_{\Omega_n}M_{n-1} + \gamma_n^2\frac{L}{2}\|V_{n+1}(X_n)\|^2\mathbf{1}_{\Omega_n}.$$

We treat each of the summands on the RHS seperately. It follows from the $G$-Lipschitz continuity and bounded variance assumption in Theorem 5.1, that

$$\mathbb{E}[\xi_{n+1}^2 \mathbf{1}_{\Omega_n}] = \mathbb{E}[\langle \nabla f(X_n), V_{n+1}(X_n) - \nabla f(X_n) \rangle^2 \mathbf{1}_{\Omega_n}]$$
$$\leq \mathbb{E}[\|\nabla f(X_n)\|^2 (\|V_{n+1}(X_n)\|^2 + 1) \mathbf{1}_{\Omega_n}] \leq G^2(C^2 + 1),$$
$$\mathbb{E}[\xi_{n+1}(1 - \gamma_n c) M_{n-1} \mathbf{1}_{\Omega_n}] = \mathbb{E}[\mathbb{E}[\xi_{n+1}|\mathcal{F}_n] M_{n-1} \mathbf{1}_{\Omega_n}] = 0,$$
$$\mathbb{E}[\|V_{n+1}(X_n)\|^2 \mathbf{1}_{\Omega_n}] \leq C. \tag{35}$$

For the term $R_{n-1} \mathbf{1}_{\tilde{E}_{n-1}}$ we have

$$\mathbb{E}[R_{n-1} \mathbf{1}_{\tilde{E}_{n-1}}] \geq \epsilon \mathbb{P}(\tilde{E}_{n-1}).$$

Using $(1 - \gamma_n c) < 1$ and putting all together we obtain the claim

$$\mathbb{E}[\tilde{R}_n] \leq \mathbb{E}[\tilde{R}_{n-1}] + \gamma_n^2 [G^2 C^2 + G^2 + C] - \epsilon \mathbb{P}(\tilde{E}_{n-1}).$$

$\square$

**Lemma D.7.** *Let $\delta > 0$ be a tolerance level and $\gamma_n \leq \gamma_1$ be sufficiently small such that $\sum_{n=1}^{\infty} \gamma_n^2 < \frac{\delta \epsilon}{2(G^2 C^2 + G^2 + C)}$ and the condition in Lemma D.6 is fulfilled. Then, we have*

$$\mathbb{P}(E_n) \geq 1 - \frac{\delta}{2}.$$

*Proof.* The proof is along the lines of the proof of Proposition D2 in Mertikopoulos et al. (2020). For completeness we repeat the arguments. First, observe that

$$\mathbb{P}(\tilde{E}_{n-1}) = \mathbb{P}(E_{n-1} \setminus E_n) = \mathbb{P}(E_{n-1} \cap \{R_n \geq \epsilon\}) = \mathbb{E}[\mathbf{1}_{E_{n-1}} \mathbf{1}_{\{R_n > \epsilon\}}] \leq \mathbb{E}[\mathbf{1}_{E_{n-1}} \frac{R_n}{\epsilon}] = \frac{\mathbb{E}[\tilde{R}_n]}{\epsilon}.$$

On the other hand it follows from Lemma D.10 that

$$\epsilon \mathbb{P}(\tilde{E}_n) \leq \mathbb{E}[\tilde{R}_n] \leq \mathbb{E}[\tilde{R}_0] + [G^2 C^2 + G^2 + C] \sum_{k=1}^{n} \gamma_k^2 - \epsilon \sum_{k=0}^{n} \mathbb{P}(\tilde{E}_{k-1}). \tag{36}$$

Rearranging everything yields

$$\sum_{k=0}^{n} \mathbb{P}(\tilde{E}_k) \leq \frac{[G^2 C^2 + G^2 + C]\Gamma}{\epsilon}$$

with $\Gamma = \sum_{n=1}^{\infty} \gamma_n^2$. By the assumption on the step size $\frac{[G^2 C^2 + G^2 + C]\Gamma}{\epsilon} < \frac{\delta}{2}$ and moreover since the events $\tilde{E}_n$ are disjoint we obtain

$$\mathbb{P}(\bigcup_{k=0}^{n} \tilde{E}_k) = \sum_{k=0}^{n} \mathbb{P}(\tilde{E}_k) \leq \frac{\delta}{2} \tag{37}$$

implying that

$$\mathbb{P}(E_n) = \mathbb{P}(\bigcap_{k=0}^{n} \tilde{E}_k^c) \geq 1 - \frac{\delta}{2}. \tag{38}$$

$\square$

**Lemma D.8.** *Let $\delta > 0$ be a tolerance level and $\gamma_n \leq \gamma_1$ be sufficiently small such that the condition in Lemma D.6 and Lemma D.7 are fulfilled. Moreover, we suppose $\gamma_1$ small enough such that $\frac{4C}{\mathbf{r}^2} \sum_{k=1}^{n} \gamma_k^2 \leq \frac{\delta}{2}$. Then, we have*

$$\mathbb{P}(\hat{E}_n) \geq 1 - \delta.$$

*Proof.* By Lemma D.8, we have $\mathbb{P}(E_n) \geq 1 - \frac{\delta}{2}$. Moreover, by the additional step size assumption and Markov's inequality we deduce that

$$
\begin{aligned}
\mathbb{P}(C_n) &= \mathbb{P}(\forall k \leq n \,:\, \|X_{k+1} - X_k\| \leq \frac{\mathbf{r}}{2}) \\
&\geq 1 - \sum_{k=1}^{n} \mathbb{P}(\|X_{k+1} - X_k\| > \frac{\mathbf{r}}{2}) \\
&= 1 - \sum_{k=1}^{n} \mathbb{P}(\|V_{k+1}(X_k)\| > \frac{\mathbf{r}}{2\gamma_k}) \\
&\geq 1 - \sum_{k=1}^{n} \mathbb{E}[\|V_{k+1}(X_k)\|^2] \frac{4\gamma_k^2}{\mathbf{r}^2} \\
&\geq 1 - \frac{4C}{\mathbf{r}^2} \sum_{k=1}^{n} \gamma_k^2 \\
&\geq 1 - \frac{\delta}{2}.
\end{aligned}
$$

Together we obtain that $\mathbb{P}(\hat{E}_n) = 1 - \mathbb{P}(\hat{E}_n^c) \geq 1 - (\mathbb{P}(E_n^c) + \mathbb{P}(C_n^c)) \geq 1 - \delta$. $\qquad\square$

Finally, we are ready to prove the main result in the local setting for the set of local minima $\mathcal{X}^*$.

*Proof of Theorem 5.1.* (i): Recall the definitions of $\mathcal{U}_1$ and $\mathcal{U}$ above. Then it holds that

$$
\Omega_{\mathcal{U}} = \bigcap_{n=1}^{\infty} \Omega_n.
$$

Hence, using Lemma D.8 we obtain

$$
\mathbb{P}(\Omega_{\mathcal{U}}) = \inf_n \mathbb{P}(\Omega_n) \geq \inf_n \mathbb{P}(\hat{E}_n) \geq 1 - \delta.
$$

(ii): We define $\tilde{D}_n := D_n \mathbf{1}_{\Omega_n}$ and prove that $\tilde{D}_n \in o(1/n^{1-\eta})$, then the claim follows since $\mathbf{1}_{\Omega_{\mathcal{U}}} \leq \mathbf{1}_{\Omega_n}$ almost surely.

From the proof of Lemma D.4 and Lemma D.6 we have

$$
\tilde{D}_{n+1} \leq \tilde{D}_n - \gamma_n c \tilde{D}_n^{2\beta} + \gamma_n \xi_{n+1} \mathbf{1}_{\Omega_n} + \frac{L\gamma_n^2}{2} \|V_n\|^2 \mathbf{1}_{\Omega_n}.
$$

Hence, taking the conditional expectation gives

$$
\begin{aligned}
\mathbb{E}[\tilde{D}_{n+1}|\mathcal{F}_n] &\leq \tilde{D}_n - \gamma_n c \tilde{D}_n^{2\beta} + \gamma_n \mathbb{E}[\xi_{n+1}|\mathcal{F}_n] \mathbf{1}_{\Omega_n} + \frac{L\gamma_n^2}{2} \mathbb{E}[\|V_{n+1}(X_n)\|^2|\mathcal{F}_n] \mathbf{1}_{\Omega_n} \\
&\leq \tilde{D}_n - \gamma_n c \tilde{D}_n^{2\beta} + \frac{LC}{2}\gamma_n^2,
\end{aligned}
$$

where we have used that $D_n$ and $\mathbf{1}_{\Omega_n}$ are $\mathcal{F}_n$-measurable and $E[\|V_{n+1}(X_n)\|^2|\mathcal{F}_n] \leq C$ from Equation (ABC) with $A = B = 0$. By our step size choice we can apply Lemma 3.1 to obtain Claim (ii).

(iii): In the following, we again separate between the two cases of $\beta$.

$\underline{\beta = \frac{1}{2}}$: We have from Lemma D.4 Equation (30) and Lemma D.6 that

$$
\tilde{D}_{n+1} \leq (1 - \gamma_n c^2)\tilde{D}_n + \gamma_n \xi_{n+1} \mathbf{1}_{\Omega_n} + \frac{L\gamma_n^2}{2} \|V_{n+1}(X_n)\|^2 \mathbf{1}_{\Omega_n}.
$$

Taking expectations and multiplying by $(n+1)^{1-\eta}$ leads to

$$\mathbb{E}[\tilde{D}_{n+1}(n+1)^{1-\eta}]$$

$$\leq (n+1)^{1-\eta}(1-\gamma_n c^2)\mathbb{E}[\tilde{D}_n] + (n+1)^{1-\eta}\frac{LC\gamma_n^2}{2}$$

$$\leq \left(n^{1-\eta} + (1-\eta)n^{-\eta}\right)(1-\gamma_n c^2)\mathbb{E}[\tilde{D}_n] + (n+1)^{1-\eta}\frac{LC\gamma_n^2}{2}$$

$$= \left(n^{1-\eta} + (1-\eta)n^{-\eta} - n^{1-\eta}\gamma_n c^2 - (1-\eta)n^{-\eta}\gamma_n c_{x^*}^2\right)\mathbb{E}[\tilde{D}_n] + (n+1)^{1-\eta}\gamma_n^2\frac{LC}{2}$$

$$= \left(1 + \frac{1-\eta}{n} - \gamma_n c^2 - \frac{(1-\eta)\gamma_n c^2}{n}\right)n^{1-\eta}\mathbb{E}[\tilde{D}_n] + (n+1)^{1-\eta}\gamma_n^2\frac{LC}{2},$$

where we used Equation (35) in the first inequality. By our choice of $\gamma_n$ there exists $\tilde{c} > 0$ and $N > 0$ such that $\gamma_n c^2 - \frac{1-\eta}{n} + \frac{(1-\eta)\gamma_n c^2}{n} \geq \tilde{c}\gamma_n$ for all $n \geq N$. Thus, for all $n \geq N$

$$w_{n+1} \leq (1 - \tilde{c}\gamma_n)w_n + (n+1)^{1-\eta}\gamma_n^2\frac{LC}{2},$$

where $w_n = \mathbb{E}[n^{1-\eta}\tilde{D}_n]$. Define $a_n = \tilde{c}\gamma_n$ and $b_n = (n+1)^{1-\eta}\gamma_n^2\frac{LC}{2}$. Since $\gamma_n = \Theta(\frac{1}{n^\theta})$, we have $\sum_n a_n = \tilde{c}\sum_n \gamma_n = \infty$ and

$$\sum_n b_n = \frac{LC}{2}\sum_n (n+1)^{1-\eta}\gamma_n^2 < \infty,$$

by Equation (24) in Lemma 3.1 Hence, we apply Lemma A.3 to prove that $\lim_{n\to\infty} w_n = 0$. By the definition of $w_n$ we have verified that $\mathbb{E}[(f(X_n) - l)\mathbf{1}_{\Omega_{\mathcal{U}}}] \leq \mathbb{E}[\tilde{D}_n] \in o(\frac{1}{n^{1-\eta}})$

$\underline{\beta \in (\frac{1}{2}, 1]}$: From Lemma D.4 Equation (31) and Lemma D.6 we have

$$\tilde{D}_{n+1} \leq (1 - \gamma_n^q c^2)\tilde{D}_n + \tilde{c}\gamma_n^{\frac{2\beta q-1}{2\beta-1}} + \gamma_n\xi_{n+1}\mathbf{1}_{\Omega_n} + \frac{L\gamma_n^2}{2}\|V_{n+1}(X_n)\|^2\mathbf{1}_{\Omega_n},$$

for $\tilde{c} = (2\beta)^{-\frac{1}{2\beta-1}}(1 - \frac{1}{2\beta})c^2$. Next we multiply with $(n+1)^{1-\eta}$ and use Equation (17) to obtain

$$\mathbb{E}[\tilde{D}_{n+1}(n+1)^{1-\eta}]$$

$$\leq (n+1)^{1-\eta}(1 - \gamma_n^q c^2)\mathbb{E}[\tilde{D}_n] + (n+1)^{1-\eta}\tilde{c}\gamma_n^{\frac{2\beta q-1}{2\beta-1}} + (n+1)^{1-\eta}\frac{LC}{2}\gamma_n^2$$

$$\leq \left(n^{1-\eta} + (1-\eta)n^{-\eta}\right)(1 - \gamma_n^q c^2)\mathbb{E}[\tilde{D}_n] + c_1(n+1)^{1-\eta}(\gamma_n^{\frac{2\beta q-1}{2\beta-1}} + \gamma_n^2)$$

$$= \left(n^{1-\eta} + (1-\eta)n^{-\eta} - \gamma_n^q c^2 n^{1-\eta} - (1-\eta)\gamma_n^q c^2 n^{-\eta}\right)\mathbb{E}[\tilde{D}_n]$$

$$\quad + c_1(n+1)^{1-\eta}(\gamma_n^{\frac{2\beta q-1}{2\beta-1}} + \gamma_n^2)$$

$$= \mathbb{E}[\tilde{D}_n n^{1-\eta}]\left(1 + \frac{1-\eta}{n} - \gamma_n^q c^2 - \frac{(1-\eta)\gamma_n^q c^2}{n}\right) + c_1(n+1)^{1-\eta}(\gamma_n^{\frac{2\beta q-1}{2\beta-1}} + \gamma_n^2),$$

for some $c_1 > 0$. By our choice of $\gamma_n$ and as $q \geq 1$, there exists a $c_2 > 0$ and $N > 0$ such that $\gamma_n^q c^2 - \frac{1-\eta}{n} + \frac{(1-\eta)\gamma_n^q c^2}{n} \geq c_2\gamma_n^q$ for all $n \geq N$. Thus, for $n \geq N$

$$\mathbb{E}[\tilde{D}_{n+1}(n+1)^{1-\eta}] \leq \mathbb{E}[\tilde{D}_n n^{1-\eta}](1 - c_2\gamma_n^q) + c_1(n+1)^{1-\eta}(\gamma_n^{\frac{2\beta q-1}{2\beta-1}} + \gamma_n^2).$$

Define $w_n = \mathbb{E}[\tilde{D}_n n^{1-\eta}]$, $a_n = c_2\gamma_n^q$ and $b_n = c_1(n+1)^{1-\eta}(\gamma_n^{\frac{2\beta q-1}{2\beta-1}} + \gamma_n^2)$. We will again apply Lemma A.3. By the step size choice $\gamma_n = \Theta(\frac{1}{n^\theta})$ we have $\sum_n a_n = c_2\sum_n \gamma_n^q = \infty$, because $q \leq \frac{1}{\theta}$. Further,

$$\sum_n b_n = c_1\sum_n (n+1)^{1-\eta}(\gamma_n^{\frac{2\beta q-1}{2\beta-1}} + \gamma_n^2) < \infty,$$

because we choose the auxiliary parameter $q$ as in the proof of Lemma 3.1 where we showed in Equation (24) and Equation (25) that

$$\sum_{n=N}^{\infty} n^{1-\eta-2\theta} < \infty \quad \text{and} \quad \sum_{n=N}^{\infty} n^{1-\eta-\frac{\theta(2\beta q-1)}{2\beta-1}} < \infty$$

All together we deduce that $w_n$ vanishes at infinity. Again, by the definition of $w_n$ we have that $\mathbb{E}[(f(X_n) - l)\mathbf{1}_{\Omega_{\mathcal{U}}}] \le \mathbb{E}[\tilde{D}_n] \in o(\frac{1}{n^{1-\eta}})$ □

## D.2 Proof of Theorem 5.2

Suppose throughout this section that the assumptions in Theorem 5.2 are satisfied.

The proof will be similar to the previous section. Instead of assuring that $(X_n)$ remains close to the set where we could guarantee the unified gradient domination property, it is now sufficient that $f(X_n)$ remains close to $f^*$ by the different definition of gradient domination definition in $f^*$. This will simplify the proof. Moreover, we may again assume w.l.o.g. the uniform second moment bounds, i.e. $A = B = 0$, instead of the more general Equation (ABC) condition by the same argument as above but on the level sets.

Recall the notation

$$\mathcal{B}_r^* = \{x \in \mathbb{R}^d \ : \ f(x) - f^* \le r\}.$$

and let $r > 0$ be the radius of the gradient domination property in $f^*$, then there exists $\epsilon > 0$, such that $2\epsilon + \sqrt{\epsilon} < r$, i.e

$$\mathcal{U} := \mathcal{B}_{2\epsilon+\sqrt{\epsilon}}^* \subset \mathcal{B}_r^*. \tag{39}$$

Moreover, we define the set

$$\mathcal{U}_1 := \mathcal{B}_{\frac{\epsilon}{2}}^* \tag{40}$$

and the measurable subsets

$$\Omega_n = \{X_k \in \mathcal{U}, \text{ for all } k \le n\}$$

in $(\Omega, \mathcal{F}, \mathbb{P})$.

The proof of Theorem 5.2 is again based on a series of auxiliary lemmas. The goal of these is to prove that with high probability we do not leave the gradient dominated region, i.e. Claim (i) in Theorem 5.2.

In the following, we fix the notation $D_n := f(X_n) - f^*$ and $\tilde{D}_n := D_n \mathbf{1}_{\Omega_n}$, $\xi_{n+1} := -\langle \nabla f(X_n), Z(X_n, \zeta_{n+1}) \rangle$ and obtain the parallel result to Lemma D.4.

**Lemma D.9.** *If $\beta = \frac{1}{2}$, it holds that*

$$D_{n+1}\mathbf{1}_{\Omega_n} \le (1 - \gamma_n c^2)D_n\mathbf{1}_{\Omega_n} + \gamma_n \xi_{n+1}\mathbf{1}_{\Omega_n} + \frac{L\gamma_n^2}{2}\mathbf{1}_{\Omega_n}\|V_{n+1}(X_n)\|^2,$$

$$\le D_1 \prod_{k=1}^{n}(1 - \gamma_k c^2)\mathbf{1}_{\Omega_n} + \sum_{k=1}^{n}\left(\prod_{j=k}^{n}(1 - \gamma_j c^2)\right)\gamma_k \xi_{k+1}\mathbf{1}_{\Omega_n} \tag{41}$$

$$+ \frac{L}{2}\sum_{k=1}^{n}\gamma_k^2\|V_{k+1}(X_k)\|^2\mathbf{1}_{\Omega_n}.$$

*If $\beta \in (\frac{1}{2}, 1]$, for any $1 \le q \le 2$, it holds that*

$$D_{n+1}\mathbf{1}_{\Omega_n} \le (1 - \gamma_n^q c^2)D_n\mathbf{1}_{\Omega_n} + (2\beta)^{-\frac{1}{2\beta-1}}(1 - \frac{1}{2\beta})c^2\gamma_n^{\frac{2\beta q-1}{2\beta-1}} + \gamma_n \xi_n \mathbf{1}_{\Omega_n} + \frac{L\gamma_n^2}{2}\|V_{n+1}(X_n)\|^2$$

$$\le D_1 \prod_{k=1}^{n}(1 - \gamma_k^q c^2)\mathbf{1}_{\Omega_n} + \tilde{c}\sum_{k=1}^{n}\gamma_k^{\frac{2\beta q-1}{2\beta-1}} + \sum_{k=1}^{n}\left(\prod_{j=k}^{n}(1 - \gamma_j^q c^2)\right)\gamma_k \xi_{k+1}\mathbf{1}_{\Omega_n} \tag{42}$$

$$+ \frac{L}{2}\sum_{k=1}^{n}\gamma_k^2\|V_{k+1}(X_k)\|^2\mathbf{1}_{\Omega_n},$$

for $\tilde{c} = (2\beta)^{-\frac{1}{2\beta-1}}(1 - \frac{1}{2\beta})c^2$.

*Proof.* The proof follows line for line as in Lemma D.4 by replacing $l$ with $f^*$ and taking the different definition of $\tilde{D}_n$ and $\Omega_n$ into account. $\square$

We continue as in the previous section:

For $\beta > \frac{1}{2}$ we know from the proof of Lemma 3.1 that we can choose the auxiliary parameter $q$ from the previous lemma in such a way, that $\sum_{n=1}^{\infty} n^{1-\eta}\gamma_n^{\frac{2\beta q-1}{2\beta-1}}$ is convergent for all $\eta \in (\max\{2 - 2\theta, \frac{\theta+2\beta-2}{2\beta-1}\}, 1)$ (Condition (iii) to apply Lemma A.2). As $\eta < 1$, it follows that $\sum_{n=1}^{\infty} \gamma_n^{\frac{2\beta q-1}{2\beta-1}} < \infty$ holds true for all these choices of $q$. Now define

$$M_n = \sum_{k=1}^{n}\left(\prod_{j=k}^{n}(1 - \gamma_j c^2)\right)\gamma_k\xi_{k+1}\mathbf{1}_{\Omega_k}, \quad M_n^{(q)} = \sum_{k=1}^{n}\left(\prod_{j=k}^{n}(1 - \gamma_j^q c^2)\right)\gamma_k\xi_{k+1}\mathbf{1}_{\Omega_k}$$

$$\text{and} \quad S_n = \frac{L}{2}\sum_{k=1}^{n}\gamma_k^2\|V_{k+1}(X_k)\|^2\mathbf{1}_{\Omega_k}.$$

Then, $(M_n)_{n\in\mathbb{N}}$ and $(M_n^{(q)})$ are $(\mathcal{F}_{n+1})$-martingales with zero mean and $(S_n)_{n\in\mathbb{N}}$ is a $(\mathcal{F}_{n+1})$-sub-martingale by Assumption 2.4. Note that by the choice of $\gamma_n$ we have $\sum_n \gamma_n^2 < \infty$ and hence $\mathbb{E}[S_n] < \infty$ for all $n \in \mathbb{N}$. Next, define $R_n = M_n^2 + S_n$ and $R_n = (M_n^{(q)})^2 + S_n$ respectively (with some abuse of notation) for every $n \in \mathbb{N}$. Moreover, let

$$E_n = \{R_k < \epsilon \text{ for all } k \le n\}.$$

which is a $\mathcal{F}_{n+1}$-measurable event on $(\Omega, \mathcal{F}, \mathbb{P})$. We define $R_0 = 0$ such that $E_0 = \Omega$.

With these definitions, we can directly prove a parallel result to Lemma D.6 without the auxiliary result in Lemma D.5.

**Lemma D.10.** *For $\beta \in (\frac{1}{2}, 1]$ let $\gamma_n \le \gamma_1$ be sufficiently small such that $\sum_{n=1}^{\infty} \gamma_n^{\frac{2\beta q-1}{2\beta-1}} < \frac{\epsilon}{2\tilde{c}}$, and for $\beta = \frac{1}{2}$ let $\gamma_1 > 0$ be arbitrary. Furthermore, assume that the initial $X_1 \in \mathcal{U}_1 = \{x : f(x) - f(x^*) \le \frac{\epsilon}{2}\}$ almost surely. Then,*

a) $E_{n+1} \subset E_n$ and $\Omega_{n+1} \subset \Omega_n$

b) $E_n \subset \Omega_{n+1}$

c) *Define the events $\tilde{E}_n = E_{n-1} \setminus E_n = E_{n-1} \cup \{R_n \ge \epsilon\}$. Then, for $\tilde{R}_n = R_n\mathbf{1}_{E_{n-1}}$, there exists a $\tilde{C} > 0$ such that*

$$\mathbb{E}[\tilde{R}_n] \le \mathbb{E}[\tilde{R}_{n-1}] + \gamma_n^2[G^2C^2 + G^2 + C] - \epsilon\mathbb{P}(\tilde{E}_{n-1}).$$

*Proof.* a) Follows by definition.

b) Note that $E_0 = \Omega = \Omega_1$ because $X_1 \in \mathcal{U}_1 = \Omega_1$ almost surely. We prove the claim by induction. Let $\omega \in E_n$. Since $E_n \subset E_{n-1} \subset \Omega_n$ by induction assumption, we have $\omega \in \Omega_n$ and thus $\omega \in \Omega_k$ for all $k \le n$. It remains to show that $X_{n+1}(\omega) \in \mathcal{U}$ to prove that $\omega \in \Omega_{n+1}$. We separate both cases for $\beta$:

$\underline{\beta = \frac{1}{2}}$: The inequality Equation (41) and the induction hypothesis yield

$$D_{n+1}(\omega) \le D_1(\omega) \prod_{k=1}^{n}(1 - \gamma_k c) + \sum_{k=1}^{n}\left(\prod_{j=k}^{n}(1 - \gamma_j c^2)\right)\gamma_k\xi_{k+1}(\omega) + \frac{L}{2}\sum_{k=1}^{n}\gamma_k^2\|V_{k+1}(X_k(\omega))\|^2$$

$$= D_1(\omega)\prod_{k=1}^{n}(1 - \gamma_k c) + \sum_{k=1}^{n}\left(\prod_{j=k}^{n}(1 - \gamma_j c^2)\right)\gamma_k\xi_{k+1}(\omega)\mathbf{1}_{\Omega_k}(\omega)$$

$$+ \frac{L}{2}\sum_{k=1}^{n}\gamma_k^2\|V_{k+1}(X_k(\omega))\|^2\mathbf{1}_{\Omega_k}(\omega)$$

$$\le \frac{\epsilon}{2} + \sqrt{R_n(\omega)} + R_n(\omega)$$

$$\le 2\epsilon + \sqrt{\epsilon}.$$

Hence, $X_{n+1}(\omega) \in \mathcal{U}$ by definition of $\mathcal{U}$.

$\underline{\beta \in (\frac{1}{2}, 1]}$: Similarly, we obtain from Equation (42)

$$D_{n+1}(\omega) \le D_1(\omega)\prod_{k=1}^{n}(1 - \gamma_k^q c) + \tilde{c}\sum_{k=1}^{n}\gamma_k^{\frac{2\beta q-1}{2\beta-1}} + \sum_{k=1}^{n}\left(\prod_{j=k}^{n}(1 - \gamma_j^q c^2)\right)\gamma_k\xi_{k+1}(\omega)$$

$$+ \frac{L}{2}\sum_{k=0}^{n}\gamma_k^2\|V_{k+1}(X_k(\omega))\|^2$$

$$= D_1(\omega)\prod_{k=1}^{n}(1 - \gamma_k^q c) + \tilde{c}\sum_{k=1}^{n}\gamma_k^{\frac{2\beta q-1}{2\beta-1}} + \sum_{k=1}^{n}\left(\prod_{j=k}^{n}(1 - \gamma_j^q c^2)\right)\gamma_k\xi_{k+1}(\omega)\mathbf{1}_{\Omega_k}(\omega)$$

$$+ \frac{L}{2}\sum_{k=0}^{n}\gamma_k^2\|V_{k+1}(X_k(\omega))\|^2\mathbf{1}_{\Omega_k}(\omega)$$

$$\le \frac{\epsilon}{2} + \frac{\epsilon}{2} + \sqrt{R_n(\omega)} + R_n(\omega)$$

$$\le 2\epsilon + \sqrt{\epsilon},$$

where we used that $\prod_{k=1}^{n}(1 - \gamma_k^q c^*) < 1$. Hence, it holds again that $X_{n+1}(\omega) \in \mathcal{U}$.

This prove that $\omega \in \Omega_{n+1}$ and closes the induction.

c) Follows line by line as in Lemma D.6 part c). $\qquad\square$

**Lemma D.11.** *Let $\delta > 0$ be a tolerance level and $\gamma_n \le \gamma_1$ be sufficiently small such that $\sum_{n=1}^{\infty}\gamma_n^2 < \frac{\delta\epsilon}{2(G^2C^2+G^2+C)}$ and the condition in Lemma D.10 is fulfilled. Then, we have*

$$\mathbb{P}(E_n) \ge 1 - \delta.$$

*Proof.* Line by line as in Lemma D.7. $\qquad\square$

Finally, we are ready to prove the main result in the local setting for $f^*$.

*Proof of Theorem 5.2.* (i): Recall the definition of $\mathcal{U}_1$ and $\mathcal{U}$ above. Then it holds that

$$\Omega_{\mathcal{U}} = \bigcap_{n=1}^{\infty}\Omega_n.$$

Hence, using Lemma D.11 we obtain

$$\mathbb{P}(\Omega_{\mathcal{U}}) = \inf_n \mathbb{P}(\Omega_n) \geq \inf_n \mathbb{P}(E_n) \geq 1 - \delta.$$

The proof of Claim (ii) and (iii) follows line by line as in the proof of Theorem 5.1 by replacing $l$ with $f^*$ and taking the different definitions of $D_n$, $\tilde{D}_n$ and $\Omega_n$ into account.

$\square$

# E  Proofs of Section 7

In the following section, we provide the proof of Section 7.

*Proof of Proposition 7.1.* We consider the cases $\lambda = 0$ and $\lambda > 0$ separately.
**Case $\lambda = 0$:** Define the optimal reward gap in every state $s \in \mathcal{S}$ by

$$\Delta^*(s) = Q^*(s, a^*(s)) - \max_{a \neq a^*(s)} Q^*(s, a) > 0,$$

where $a^*(s)$ denotes the best possible action in state $s$ and $Q^* : \mathcal{S} \times \mathcal{A} \to \mathbb{R}$ denotes the optimal Q-function defined by $Q^*(s, a) = \mathbb{E}_\mu^{\pi^*}[\sum_{t=0}^\infty \rho^{-t} r(S_t, A_t)|A_0 = a]$. W.l.o.g. we assume that $a^*(s)$ is unique. Similarly let $Q^{\pi_w}(s, a) = \mathbb{E}_\mu^{\pi_w}[\sum_{t=0}^\infty \rho^{-t} r(S_t, A_t)|A_0 = a]$ be the Q-function for policy $\pi_w$.

For any $0 < \alpha < 1$ choose $r = \min_{s \in \mathcal{S}} \mu(s) \min_{s \in \mathcal{S}} \Delta^*(s)(1-\alpha)$ and assume that $w \in \mathcal{B}_r^*$, i.e. $V^*(\mu) - V^{\pi_w}(\mu) \leq r$. Then, we have for every $s \in \mathcal{S}$ that

$$V^*(\delta_s) - V^{\pi_w}(\delta_s) \leq \frac{r}{\min_{s \in \mathcal{S}} \mu(s)}.$$

It follows for every $s \in \mathcal{S}$ that

$$\frac{r}{\min_{s \in \mathcal{S}} \mu(s)} \geq V^*(\delta_s) - V^{\pi_w}(\delta_s)$$

$$= Q^*(s, a^*(s)) - \sum_{a \in \mathcal{A}_s} \pi_w(a|s) Q^{\pi_w}(s, a)$$

$$\geq \sum_{a \in \mathcal{A}_s} \pi_w(a|s)(Q^*(s, a^*(s)) - Q^*(s, a))$$

$$= \sum_{a \neq a^*(s)} \pi_w(a|s)(Q^*(s, a^*(s)) - Q^*(s, a))$$

$$\geq (1 - \pi_w(a^*(s)|s)) \min_s \Delta^*(s).$$

Rearranging results in

$$\pi_w(a^*(s)|s) \geq 1 - \frac{r}{\min_{s \in \mathcal{S}} \mu(s) \min_{s \in \mathcal{S}} \Delta^*(s)} = \alpha.$$

Hence, for all $w \in \mathcal{B}_r^*$ we can bound $c(w)$ by

$$c(w) \geq \frac{\alpha}{\sqrt{|\mathcal{S}|}(1-\rho)} \left\| \frac{d_\mu^{\pi^*}}{\mu} \right\|_\infty^{-1} > 0.$$

Thus, setting $c = \frac{\alpha}{\sqrt{|\mathcal{S}|}(1-\rho)} \left\| \frac{d_\mu^{\pi^*}}{\mu} \right\|_\infty^{-1}$ proves the claim.

**Case** $\lambda > 0$**:** For any $\alpha \in (0,1)$ choose $r = \alpha^2 \exp\left(\frac{-1}{(1-\rho)\lambda}\right)^2 \frac{\lambda \min_s \mu(s)}{2\ln 2}$ and assume that $w \in \mathcal{B}^*_{r,\lambda}$. By Ding et al. (2023, Lem. 12) we have

$$
\begin{aligned}
|\pi_w(a|s) - \pi^*(a|s)| &\leq \sqrt{\frac{2(V^*_\lambda(\mu) - V^{\pi_w}_\lambda(\mu))\ln 2}{\lambda \min_s \mu(s)}} \\
&\leq \sqrt{\frac{2r\ln 2}{\lambda \min_s \mu(s)}} \\
&= \alpha \exp\left(\frac{-1}{(1-\rho)\lambda}\right) \\
&\leq \alpha \min_{s,a} \pi^*(a|s).
\end{aligned}
$$

where the last inequality is due to Nachum et al. (2017, Thm. 1). It follows directly that

$$
\min_{s,a} \pi_w(s,a) \geq (1-\alpha) \min_{s,a} \pi^*(s,a) > 0.
$$

Hence, we can bound $c(w)$ uniformly for all $w \in \mathcal{B}^*_{r,\lambda}$ by

$$
c(w) \geq \frac{2\lambda}{|\mathcal{S}|(1-\rho)} \min_s \mu(s)(1-\alpha)^2 \min_{s,a} \pi^*(a|s)^2 \left\| \frac{d^{\pi^*}_\mu}{\mu} \right\|^{-1}_\infty. \tag{43}
$$

Thus, setting $c = \frac{2\lambda}{|\mathcal{S}|(1-\rho)} \min_s \mu(s)(1-\alpha)^2 \min_{s,a} \pi^*(a|s)^2 \left\| \frac{d^{\pi^*}_\mu}{\mu} \right\|^{-1}_\infty$ proves the claim. $\qquad \square$

*Remark* E.1. It is noteworthy, that we have multiple choices of $r$ and $c$ depending on $\alpha \in (0,1)$.

