# OpenReview forum: "Almost Sure Convergence of Stochastic Gradient Methods under Gradient Domination"
_TMLR — Accepted by TMLR_

### Review · Reviewer_CDgH · 2025-02-03

**Summary Of Contributions:**

This paper studies almost-sure convergence rates for stochastic gradient descent (SGD) under gradient domination, the condition that $\| \nabla f(x) \| \geq (f(x) - f_{\ast})^{2\beta}$, where $\beta \in [\frac{1}{2}, 1]$. This condition is a generalization of the well-known PL inequality, and the authors consider both a global version of it (where the assumption holds over all of $\mathbb{R}^d$, the domain of $f$), and two different local versions (one defined by the assumption holding over a ball around the minimizer, and one defined by the assumption holding over a sublevel set). Theorem 4.1 gives the first main result of the paper: the almost-sure convergence of SGD under gradient domination, alongside convergence in expectation (though the latter is not new). Theorem 4.2 gives a similar result for the stochastic heavy-ball method. Unfortunately, the rate there is no better than that of SGD but this is to be expected, as optimization under the PL condition seems to be strictly harder than optimization under strong convexity, in the sense that it cannot be accelerated (Yue et al., 2023). In any case, this is the stochastic setting, so any improvement from acceleration would be in higher-order terms and asymptotically negligible anyway. Theorems 5.1 and 5.2 give similar results but under the local versions of the condition.

Yue, Pengyun, Cong Fang, and Zhouchen Lin. "On the lower bound of minimizing polyak-Łojasiewicz functions." The Thirty Sixth Annual Conference on Learning Theory. PMLR, 2023.

**Audience:**

Yes

**Broader Impact Concerns:**

N/A.

**Claims And Evidence:**

Yes

**Requested Changes:**

- Can you elaborate on this "Provided that σ and Φ are analytic, and (Z, Y ) are compactly supported Rdz × Rdy -valued random variables,
then f DNN : Rdw → R+ defined by w 7 → Eμ(Z,Y ) [Φ(gw(Z), Y )] is analytic (Dereich and Kassing, 2024, Thm. 5.2) and therefore satisfies local gradient domination in any stationary point w∗ (see Definition 2.2) Lojasiewicz (1965)." What would be the local gradient domination constant in this case? Can you give a more precise argument?
- In p. 12, the "The proof of this proposition is given in ??" should point to p. 35/36. p.35 "In the following section, we provide the proof of ??." should be fixed.
- Appendix D doesn't show up in the table of contents for some reason, but the other appendices do.

**Strengths And Weaknesses:**

- The paper is clear and well-written. The arguments are explained in a way that encourages broader adoption.
- Almost-sure convergence rates are relatively rare, this is a welcome improvement! While the asymptotic nature of these guarantees makes them somewhat weaker than non-asymptotic bounds, they still provide value. A drawback of the current work is that there are no non-asymptotic high-probability rates presented. These would be very beneficial and seem like they might follow using similar probabilistic tools as to those used in the paper.
- The noise assumption used is very general. It allows for affine noise, and is only assumed to hold in expectation. It's quite nice that almost-sure convergence can be obtained despite assuming the noise assumption holds only in expectation.
-  I like the application to reinforcement learning, it is an interesting example of a more interesting example-- even though it only applies to finite state spaces and the domination constant can be really really tiny (it scales inversely with the size of the state space..)
- The local gradient domination results require somewhat strong assumptions about remaining in the "good" region. The stepsize $\gamma_1$ can be very, very tiny based on the required condition (the sum in Lemma E.6. being bounded). More discussion of this might strengthen the paper.

---

> ### Author Response · Authors · 2025-02-13
> **Response to Reviewer CDgH**
>
> Thank you very much for taking the time to review our article! In the following, we give detailed responses to each points.
>
> **Strengths And Weaknesses:**
> >  Almost-sure convergence rates are relatively rare, this is a welcome improvement! While the asymptotic nature of these guarantees makes them somewhat weaker than non-asymptotic bounds, they still provide value. A drawback of the current work is that there are no non-asymptotic high-probability rates presented. These would be very beneficial and seem like they might follow using similar probabilistic tools as to those used in the paper.
>
> We agree that it would be intrigue to derive non-asymptotic high-probability rates based on our probabilistic tools applied in the paper. However, this extension would be beyond the scope of the paper and we leave this question for future work.
>
> > The local gradient domination results require somewhat strong assumptions about remaining in the "good" region. The stepsize  can be very, very tiny based on the required condition (the sum in Lemma E.6. being bounded). More discussion of this might strengthen the paper.
>
> Thank you for bringing up this point. We have included the explicit requirements on $\gamma_n$ in the main body of our paper and added a discussion about the dependence of $\gamma_n$ on $\delta$.
>
> **Requested Changes:**
>
> > Can you elaborate on this "Provided that $\sigma$ and $\Phi$ are analytic, and $(Z,Y)$ are compactly supported $R^{dz} \times R^{dy}$ -valued random variables, then $f_{DNN}: R^{dw} \to R_+$ defined by $w \to E_{\mu_{(Z,Y)}} [\Phi(g_w(Z),Y)]$ is analytic (Dereich and Kassing, 2024, Thm. 5.2) and therefore satisfies local gradient domination in any stationary point $w_\ast$ (see Definition 2.2) Lojasiewicz (1965)." What would be the local gradient domination constant in this case? Can you give a more precise argument?
>
> Thank you for the suggestion. We have refined the statement to be more precise and now explicitly clarify that for an analytic function, at any stationary point $w_\ast$, there exist two constants $\beta_{w_\ast}\in[1/2,1]$ and $c_{w_\ast}>0$ such that the local gradient domination property holds. However, this statement does not provide an explicit characterization of these constants.
>
> > In p. 12, the "The proof of this proposition is given in ??" should point to p. 35/36. p.35 "In the following section, we provide the proof of ??." should be fixed.
>
> Thank you, we have corrected it.
>
> > Appendix D doesn't show up in the table of contents for some reason, but the other appendices do.
>
> We have incorporated the proofs from Appendix D of the previous version into the main body of our revised manuscript. As a result, this appendix no longer exists, and the issue should now be resolved.

---

> > ### Comment · Reviewer_CDgH · 2025-02-18
> >
> > Thank you for addressing my points. I have no more issues and recommend the paper be accepted.

---

### Review · Reviewer_euVb · 2025-02-03

**Summary Of Contributions:**

The paper offers a new set of results on stochastic optimization under gradient dominated condition, important generalization of well-known Polyak-Lojasievicz inequality. The key results of this work are the extension of in-expectation result to almost sure convergence guarantee, and the study of SGD (and heavy ball momentum) under local variant of gradient dominated condition. It is also demonstrated how the new theory can be applied to training neural networks and policy gradient methods in reinforcement learning.

**Audience:**

Yes

**Claims And Evidence:**

Yes

**Requested Changes:**

Please include the precise dependence on failure probability in the statement of main theorems 5.1 and 5.2., and consider the suggestions above.

**Strengths And Weaknesses:**

**Strengths:**
1. The paper derives interesting results with almost sure convergence. The results with this strong convergence type are in the literature, and can be difficult to establish?

2. The paper is well-written with a clear positioning of the contributions. The literature review seems comprehensive.  The motivation for the study is convincing.

**Weakness:**
1. **The main issue is that the claim of the main theorems 5.1 and 5.2 are not meaningful unless the dependence on failure probability $\delta$ is specified in the rate.**
2. It is also important to specify dependence on other constants. Specifically, the dependence on gradient dominance constant is important as it can be very small even for very simple non-convex problems, see, e.g., (Jarre, 2011).

**Suggestion for improvements:**
1. it is better to discuss separately the regularized and unregularized settings after Corollary 7.3., and carefully compare to prior work in these settings in both deterministic and stochastic settings. There are some works which derived similar local rates under weak gradient dominance and local hidden convexity (Junyu Zhang et al., 2021, Barakat et al., 2023).
2. It would be useful to comment on what quantities are plotted in bold and not bold in toy experiment in appendix B. I assume these are all runs and the median/mean in bold?
3. Is it possible to discuss the rates in more detail in the special cases $\beta=1/2$ and $\beta=1$? Is there any "practical" interest in the study of the values in between, i.e., are there any problems which satisfy your assumption with $\beta \in (1/2, 1)$ but not for $\beta = 1/2$?


**Typos and inaccuracies:**
1. "step size is is small".
2. Should be $f$? "In Section 7 we will show how this non-uniform gradient domination implies local gradient domination for $f^∗$."
3. On page 12, the reference is missing: "The proof of this proposition is given in ??."
4. What does Remark 7.2. mean? What is $\alpha$?
5. Missing reference on page 35: "In the following section, we provide the proof of ??."
6. This sentence is not accurate because works do not fit the description in the sentence. The former work does not consider neural networks. The latter does not prove 1/T rate. "See also Madden et al. (2021) and Liu et al. (2022), where convergence rates of order O( 1 / T ) are shown for neural networks using the (strong) gradient domination property".
7. This description is also not accurate because Liu and Yuan (2022) consider PL inequality, not only strong convexity: "In recent years, Sebbouh et al. (2021) and, building upon it, Liu and Yuan (2022) derive almost sure convergence rates towards global optima under strong convexity."

**Questions:**
1. Why is this the case? "If λ > 0 there exists a continuum of  optimal parameters w∗, such that V πw∗  λ (μ) = V ∗  λ (μ)."


Florian Jarre, On Nesterov’s Smooth Chebyshev-Rosenbrock Function. 2011.
Junyu Zhang et al. On the convergence and sample efficiency of variance-reduced policy gradient method. NeurIPS 2021.
Anas Barakat et al. Reinforcement Learning with General Utilities: Simpler Variance Reduction and Large State-Action Space. ICML 2023.

---

> ### Author Response · Authors · 2025-02-13
> **Response to Reviewer euVB**
>
> Thank you very much for taking the time to review our article! In the following, we give detailed responses to each points.
>
> **Weakness:**
>
> 1. Thank you very much for your valuable critique. We have now explicitly incorporated the requirement on $(\gamma_n)$ in terms of $\delta$ within our theorems. Notably, this is the only aspect of our convergence result that depends on $\delta$. In the original version of the manuscript, these details were provided within the appendix and we agree that they should be presented more transparently in the main theorem. \
> The potential confusion may arise from comparisons to standard "high probability" convergence results, where the error is shown to converge in probability with a sample complexity that explicitly depends on $\delta$. However, we want to emphasize that we do not provide convergence rates in high probability. Instead, we restrict the probability space to "good" events that occur with high probability. Conditioned on these events, we establish convergence with a rate that is independent of $\delta$. That said, we acknowledge that our convergence results are asymptotic in time. We do not provide non-asymptotic bounds, where the dependence on $\delta$ would indeed play a crucial role. For a clearer understanding of the theorem, we have included a discussion behind Theorem 5.1 in the revised manuscript, which further highlights this distinction.
>
> 2. We agree that an explicit dependence on other constants would be desirable and certainly an interesting direction for future work. In this study, however, the convergence rates are based on the Robbins-Siegmund result and are asymptotic in nature, which precludes the possibility of explicitly expressing their dependence on the constants.
>
> **Suggestion for improvements:**
> 1. Thank you for pointing us to the two references. We recognize that there are many other valuable works on the convergence of (regularized) policy gradient methods. However, this section of the paper is primarily intended to illustrate an application of Theorem 5.2. We focused on literature of stochastic gradient methods and opted for a compact presentation of the RL part, considering both the regularized and regular cases together. We added a brief discussion on the related work to our proved convergence result after Corollary 7.2.
>
> 2. Thank you for pointing us to this issue, we now clarify in the caption of Figure 1 that the bold lines correspond to the average over the 100 runs.
>
> 3. Indeed, all analytic functions satisfy the local {\L}ojasiewicz condition for some $\beta\in[\frac 12,1]$. Therefore, the application to analytic neural networks represents a case where the general scenario for $\beta$ is required. In Section 6 of the revised manuscript, we have explicitly emphasized the general dependence on $\beta$ to make this point clearer.
>
> **Typos and inaccuracies:**
>
> Thank you for your careful reading. We have corrected all typos and inaccuracies. Moreover, we give the following comments:
>
> 4. Since $\alpha$ occurs in the proof of Proposition 7.1., we have decided to move Remark 7.2. into the appendix behind the proof of Proposition 7.1.
>
> 6. Thank you again for pointing out the inaccurate wording. We were unaware of the updated version of Madden et al. (2021) and have revised the sentence accordingly.
>
>
> 7. Liu and Yuan (2022) assume strong convexity, but ultimately rely on the PL inequality in their proofs. Therefore, we have included their work in our overview table. Thank you for this comment; we updated the sentence accordingly.
>
> **Questions:**
>
> 1. The existence of a continuum of optimal parameters has been discussed in Ding et al. (2022). Following Nachum et al. (2017), in the entropy-regularized setting, there exists a unique optimal policy represented by the softmax parametrization. However, due to the structure of the softmax parameterization, there is an entire continuum of parameter values that represent the same policy. We have clarified this point in our revised manuscript by adding both references.

---

> ### Comment · Reviewer_euVb · 2025-02-19
> **Mistakes in Equations (14) and (15)**
>
> I am afraid there could be some mistakes in the derivations of eq. (14) and (15) of the updated manuscript.
>
> 1. Could you please check the dimensionality of the second term of eq. (14) on the RHS? It seems to me some constant is missing in front of $\frac{1}{2} \||\nabla f(X_n)\||^2$. Probably this mistake is inherited from the prior work Liu and Yuan (2022).
>
> 2. Even if equation (14) is correct, there is a mistake in how equation (15) is derived from (14) using (13). It should be $(1 - L) (f(X_n) - f^*)$ provided that (14) is correct. BTW, the bracket is also missing here.

---

> > ### Author Response · Authors · 2025-02-20
> > **Correction of Equations (14) and (15)**
> >
> > Thank you again for your careful reading of our proofs. You are absolutely right, there is a typo in equation (14) where $||\nabla f(Z_n)||^2$ should appear instead of $||\nabla f(X_n)||^2$. The goal is to establish convergence of $f(X_n)-f^*$, for which we derive an upper bound in terms of $Q_n+||W_n||^2$. We have now corrected both equations (14) and (15) accordingly.

---

> > > ### Comment · Reviewer_euVb · 2025-02-21
> > > **There is still a problem in equation (14)**
> > >
> > > I think there is still a problem with equation (14) since the dimensionality/units of the LHS and the second term on the RHS do not match. To put it simply, if function value $f$ measures units [meters] and variable x is [seconds], then the RHS is [meters], but the second term in the RHS is $\||\nabla f(Z_n)\|^2$ which is [meters^2/seconds^2]. This may imply that there is an issue with equation (14), which was borrowed from prior work.

---

> > > > ### Author Response · Authors · 2025-02-23
> > > > **Clarification of equation (14)**
> > > >
> > > > Thanks again for your careful consideration. Let us first clarify the derivation of equation (14): Recall that $Z_n = X_n + \frac{\nu}{1-\nu}W_n$. The descent condition states that for any $x,y\in\mathbb R$ we have
> > > > $$f(y)\le f(x) + \langle \nabla f(x),y-x\rangle + \frac{L}{2}||y-x||^2 $$
> > > > which follows from $L$-smoothness of $f$; see, for instance, Lemma 1.2.3 in Nesterov's lecture notes (2004). Now, settting $y = X_n$ and $x = Z_n$ we obtain
> > > > $$f(X_n) \le f(Z_n) + \langle \nabla f(Z_n),X_n-Z_n\rangle + \frac{L}2 ||X_n-Z_n||^2 = f(Z_n) - \frac{\nu}{1-\nu}\langle \nabla f(Z_n),W_n\rangle + \frac{L\nu^2}{2(1-\nu)^2} ||W_n||^2.$$
> > > > Next, apply Cauchy-Schwarz and Young's inequality to obtain
> > > > $$f(X_n)-f^* \le f(Z_n)-f^* + \frac{1}{2}||\nabla f(Z_n)||^2 + \frac{\nu^2}{2(1-\nu)^2}||W_n||^2 + \frac{L\nu^2}{2(1-\nu)^2} ||W_n||^2. $$
> > > > This expression coincides with equation (14).
> > > >
> > > > Regarding the confusion about dimensionality/units:
> > > > Note that the right-hand side of equation (14) is real-valued (and even non-negative). However, care must be taken when incorporating units.
> > > > For example, if we formulate the gradient descent scheme (or its stochastic variant) as
> > > > $$ x_{n+1} = x_n + \alpha_n \nabla f(x_n)$$
> > > > the step size $\alpha_n$ ensures that $\alpha_n \nabla f(x_n)$ is measured in $[seconds]$. That said, when introducing units into equation (14) different physical dimensions must be considered. Specifically, in the descent condition
> > > > $$f(y)\le f(x) + \langle \nabla f(x),y-x\rangle + \frac{L}{2}||y-x||^2, $$
> > > > the rhs is measured in $[meters]$, where $L$ is in $[meters/seconds^2]$ when viewed as upper bound on the Hessian. Importantly, applying Young’s inequality in the derivation of equation (14) does not alter the units, only the numerical values; $[meters/seconds\cdot seconds]$ is not bounded by $\frac12 [meters^2/seconds^2] + \frac12 [seconds^2]$.

---

> > > > > ### Comment · Reviewer_euVb · 2025-02-24
> > > > >
> > > > > Thank you for the explanation, I have no further questions.

---

### Review · Reviewer_h3cd · 2025-02-04

**Summary Of Contributions:**

The authors studied the convergence rate of stochastic gradient descent (SGD) and stochastic heavy ball (SHB) under the global/local gradient dominant property. In particular, for SGD, under the global dominant property, the proposed upper bound is arbitrarily close to the upper bound in the expectation given in (Fatkhullin et al. 2022). Moreover, they provided additional results for SGD/SHB under global/local dominant property in both almost sure and expectation guarantees. In the proofs, they used Robbins-Siegmund theorem in order to derive the convergence rates.

**Audience:**

Yes

**Broader Impact Concerns:**

Nothing specific to be mentioned.

**Claims And Evidence:**

Yes

**Requested Changes:**

- Please provide a proof sketch of Theorem 4.1 in the main body and mention any technical novelty in the proof compared to previous work.
- As far as I know, the results for almost sure guarantees in previous work require that the norm of gradient is bounded almost surely. However, it seems that Assumption 4.2 is in expectation. I think it is better to emphasize this in the paper and compare the assumptions on the stochastic gradients in Table 1.
- Regarding the tightness of the bound in the expectation in (Fatkhullin et al. 2022), to my understanding, they did not provide a lower bound. There, it is mentioned that if there is a function $f$ and a gradient estimator such that recursive inequality becomes equality, then their bound is tight. However, the main challenge is whether such a function and gradient estimator can be constructed to satisfy all the considered assumptions. I would suggest checking the following work on deriving lower bounds on the convergence rate under gradient dominant property: https://arxiv.org/pdf/2408.01839

**Strengths And Weaknesses:**

Strengths:
- Deriving the convergence rate for SGD and SHB under both local and global gradient dominant properties
- Providing almost sure bounds (to the best of my knowledge, they are novel) for SGD/SHB under local/global dominant properties
- Applying the results in two settings of deep neural networks in supervised learning and policy gradient methods in reinforcement learning

Weakness:
- The main weakness is that no proof sketch of the main theorems is given in the main body.

---

> ### Author Response · Authors · 2025-02-13
> **Response to Reviewer h3cd**
>
> Thank you very much for taking the time to review our article! In the following, we give detailed responses to each points.
>
> **Strength and Weakness:**
>
> > The main weakness is that no proof sketch of the main theorems is given in the main body.
> > Please provide a proof sketch of Theorem 4.1 in the main body and mention any technical novelty in the proof compared to previous work.
>
> We agree that the proofs of Theorem 4.1 and 4.2 should be part of the main paper. Since a sketch would essentially describe each line of the proof in words, we have chosen to present the full proof in the main text. The main technical challenge is Moreover, we have added a sketch of the proof of Theorem 5.1, Theorem 5.2 follows the same outline which we have omitted in the main body.
>
> Regarding the technical novelty: We acknowledge that using the Robbins-Siegmund Theorem to prove almost sure convergence (without rates) is standard practice. However, its application to derive a convergence rate was first introduced in Seebouh et al. (2021) and further explored by Liu & Yuan (2022). To our knowledge, deriving almost sure convergence rates is a recent topic of interest and is not yet considered standard. More specifically, in our considered setting it is a non-trivial generalisation to derive convergence rates by assuming gradient domination compared to strong convexity in Liu & Yuan (2022) as one obtains a weaker sub-martingale form given in our Lemma 3.1.
>
> **Requested Changes:**
>
> > As far as I know, the results for almost sure guarantees in previous work require that the norm of gradient is bounded almost surely. However, it seems that Assumption 4.2 is in expectation. I think it is better to emphasize this in the paper and compare the assumptions on the stochastic gradients in Table 1.
>
> In Section 3, we have now added a comment regarding the noise assumption in (Sebbouh et al., 2021) and (Liu & Yuan, 2022). Both of these works assume a similar ABC condition on the stochastic gradients as we do in Assumption 2.4.
>
> The fact that the almost sure convergence results holds under noise assumptions in expectation is a consequence of the factorization property of the conditional expectation. By conditioning on the canonical filtration, we can directly apply the ABC condition, bypassing the need of almost sure requirements on the noise. Note that this step is fairly standard in the convergence analysis of SGD and often suppressed in the literature.
>
> > Regarding the tightness of the bound in the expectation in (Fatkhullin et al. 2022), to my understanding, they did not provide a lower bound. There, it is mentioned that if there is a function  and a gradient estimator such that recursive inequality becomes equality, then their bound is tight. However, the main challenge is whether such a function and gradient estimator can be constructed to satisfy all the considered assumptions. I would suggest checking the following work on deriving lower bounds on the convergence rate under gradient dominant property: https://arxiv.org/pdf/2408.01839
>
> Thank you for bringing up this point. We agree that the results of Fatkhullin et al. (2022) do not provide an explicit lower bound. We have adjusted the formulation accordingly. Additionally, we appreciate the reference to lower bounds under gradient domination, which we have now incorporated into our literature review.

---

> > ### Comment · Reviewer_h3cd · 2025-02-14
> >
> > Thank you for addressing my comments. It might be good to add a remark about technical novelty to the paper. I have no further comments.

---

### Decision · Action_Editor_9CAP · 2025-03-07

**Recommendation:** Accept as is

**Comment:**

The paper received generally positive reviews, with reviewers highlighting its contributions to the analysis of almost sure convergence in stochastic optimization.
The two main issues raised during the review process were:
- the lack of a proof sketch for the main theorems in the initial submission
- need for a clearer discussion on noise assumptions and dependence on $\delta$.

This was addressed in the revision during the rebuttal.

Given the strength of the theoretical contributions, the careful revisions made by the authors, and the reviewers’ comments on the paper’s significance, I recommend acceptance.

**Audience:**

The results presented in this paper are of interest to the TMLR audience, for the theoretical inclined.

**Claims And Evidence:**

The paper provides results regarding the almost sure convergence of stochastic gradient descent (SGD) and the stochastic heavy-ball (SHB) method under global and local gradient dominant conditions. The authors use the Robbins-Siegmund theorem to derive convergence rates, and their results extend known expectations-based bounds. The inclusion of almost sure bounds, particularly under the local gradient

Reviewers noted the strength of these contributions and found the theoretical justifications convincing, especially after the rebuttal phase.